# When Visual Prompt Tuning Meets Source-Free Domain Adaptive Semantic Segmentation

**Xinhong Ma, Yiming Wang, Hao Liu, Tianyu Guo, Yunhe Wang**[*]
Huawei Noah's Ark Lab
{maxinhong, wangyiming22, liuhao296, tianyu.guo, yunhe.wang}@huawei.com

## Abstract

Source-free domain adaptive semantic segmentation aims to adapt a pre-trained source model to the unlabeled target domain without accessing the private source data. Previous methods usually fine-tune the entire network, which suffers from expensive parameter tuning. To avoid this problem, we propose to utilize visual prompt tuning for parameter-efficient adaptation. However, the existing visual prompt tuning methods are unsuitable for source-free domain adaptive semantic segmentation due to the following two reasons: (1) Commonly used visual prompts like input tokens or pixel-level perturbations cannot reliably learn informative knowledge beneficial for semantic segmentation. (2) Visual prompts require sufficient labeled data to fill the gap between the pre-trained model and downstream tasks. To alleviate these problems, we propose a universal unsupervised visual prompt tuning (Uni-UVPT) framework, which is applicable to various transformer-based backbones. Specifically, we first divide the source pre-trained backbone with frozen parameters into multiple stages, and propose a lightweight prompt adapter for progressively encoding informative knowledge into prompts and enhancing the generalization of target features between adjacent backbone stages. Cooperatively, a novel adaptive pseudo-label correction strategy with a multiscale consistency loss is designed to alleviate the negative effect of target samples with noisy pseudo labels and raise the capacity of visual prompts to spatial perturbations. Extensive experiments demonstrate that Uni-UVPT achieves state-of-the-art performance on GTA5 $\to$ Cityscapes and SYNTHIA $\to$ Cityscapes tasks and can serve as a universal and parameter-efficient framework for large-model unsupervised knowledge transfer. Code will be available at `https://gitee.com/mindspore/models/tree/master/research/cv/uni-uvpt` and `https://github.com/huawei-noah/noah-research/tree/master/uni-uvpt`.

## 1 Introduction

Semantic segmentation is a critical computer vision task, which aims to segment and parse a scene image into different image regions associated with semantic categories. The success of these semantic segmentation techniques relies on large-scale densely-labeled datasets. However, it is prohibitively expensive to collect high-quality annotations. Besides, most semantic segmentation methods ignore the distribution shift between training and testing data, making them fail to generalize when deployed in conditions different from training such as cross-city [8] or cross-weather [39] scenarios. *Unsupervised Domain Adaptation (UDA)* [11] is an intuitive direction to address the above two issues by transferring knowledge from existing models trained on source datasets to the unlabeled target domain. Typical UDA approaches require joint access to both labeled source and unlabeled target data during training, making them unsuitable for *Source-Free Unsupervised Domain*

---

[*]Corresponding author

37th Conference on Neural Information Processing Systems (NeurIPS 2023).

*Adaptation (SFUDA)* where source and target data are not concurrently accessible. This SFUDA setting with restricted data-sharing is privacy-oriented and thus holds immense practical value in autonomous driving and other commercial visual applications.

With the above considerations, this paper proposes to solve source-free domain adaptive semantic segmentation task, where only source pre-trained models and unlabeled target data are available for model adaptation. Recently, a few source-free UDA methods have been developed to tackle semantic segmentation [30, 19, 20, 46] and image classification [21, 23, 26]. Most of them either perform self-training [30, 1, 27, 44] or synthesize target-style training data [35] to fine-tune all model parameters. However, fine-tuning requires many computation resources, which is an expensive and often infeasible proposition for previous SFUDA methods, especially for modern transformer-based architectures. Recently, prompt tuning has been proposed to explore the knowledge of frozen language models [22, 25, 28, 29], which could be an alternative strategy to efficiently adapt the pre-trained source model to the target domain.

*Prompt Tuning* aims to design a trainable lightweight block as a supplementary input (*prompt*) for a frozen model, which could guide or direct the generalization of powerful representations to achieve desirable performances. Inspired by its great success in natural language processing (NLP) [28], a few methods in computer vision propose to add small amounts of learnable parameters as tokens [18] or pixel-level perturbations [4] to adapt large pre-trained models to downstream tasks. However, the existing visual prompt tuning methods have two limitations when applied to source-free domain adaptive semantic segmentation: (1) The learned visual prompts are unreasonable. Specifically, prompts like input tokens or pixel-level perturbations are black boxes, which cannot reliably explore convincing knowledge beneficial for pixel-wise predictions. Besides, most methods utilize task-oriented loss for optimization, *e.g.*, cross-entropy loss, which limits the learning capacities of visual prompts. (2) Previous methods cannot be directly applied to the unlabeled target domain because most of them rely on sufficient labeled data to deal with the gap between pre-trained models and downstream tasks. There is a lack of methods addressing unsupervised visual prompt tuning where only massive unlabeled data is available for training.

To address the above problems, we propose a *Universal Unsupervised Visual Prompt Tuning (Uni-UVPT)* framework for source-free domain adaptive semantic segmentation. Specifically, given a source pre-trained model, we partition its backbone with frozen parameters into multiple stages and design a lightweight prompt adapter for stage-wise prompt tuning. The prompt adapter contains a prompt generator and several prompt interactors. The former aims to generate reasonable prompts that capture multiscale spatial information and task-shared knowledge. The latter progressively refines prompts between adjacent stages and transforms target features to match the source pre-trained knowledge in the backbone. To learn prompts with massive unlabeled target samples, we utilize target instances with pseudo labels for self-training, and propose an adaptive pseudo-label correction strategy to alleviate pseudo-label noises. The proposed adaptive pseudo-label correction strategy could determine suitable moments to rectify pseudo labels and guide the model to train on instances with corrected pseudo labels, not noisy ones. Meanwhile, a multiscale consistency loss is proposed to impose multiscale consistency of the features and predictions so that the learned visual prompts are robust to arbitrary semantic-preserving spatial perturbations. It is noticed that our method is applicable to various transformer architectures without modifying their basic units.

The contributions of this work are summarized as follows: (1) We first highlight the low-efficiency problem of fine-tuning large-scale backbones in source-free domain adaptive semantic segmentation, and propose a universal unsupervised visual prompt tuning framework for parameter-efficient model adaptation. (2) A lightweight prompt adapter is introduced to learn reasonable visual prompts and enhance feature generalization in a progressive manner. Cooperatively, a novel adaptive pseudo-label correction strategy is proposed to rectify target pseudo labels at suitable moments and improve the learning capacity of visual prompts. Extensive experimental results demonstrate that our method with a few trainable parameters could achieve state-of-the-art performances on GTA5 $\rightarrow$ Cityscapes and SYNTHIA $\rightarrow$ Cityscapes tasks.

## 2   Universal Unsupervised Visual Prompt Tuning

In unsupervised domain adaptation, a labeled source dataset $\mathcal{D}_s = \{(x_s, y_s) : x_s \in \mathcal{X}_s, y_s \in \mathcal{C}\}$ and an unlabeled target dataset $\mathcal{D}_t = \{x_t : x_t \in \mathcal{X}\}$ are available for training, where $\mathcal{X}$ is the input space

and $\mathcal{C}$ denotes the label set. $x_s$ is drawn from the marginal distribution $p_s$ while $x_t$ is drawn from another marginal distribution $p_t$. The goal is to learn a mapping $f : \mathcal{X} \rightarrow \mathcal{C}$ that can generalize well for $x_t$. We focus on the source-free constraint [30], where we do not have access to the original source domain data $\mathcal{D}_s$. Instead, merely a source pre-trained model $f_s$ and unlabeled target data $\mathcal{D}_t$ are provided for adaptation. The source pre-trained semantic segmentation model usually contains a backbone $f_b$ (containing some embedding layers, *e.g.*, the patch embedding layer in Swin [31]) and a customized head $f_h$. To efficiently adapt the source pre-trained model $f_s$ to the unlabeled target domain $\mathcal{D}_t$, we propose to perform unsupervised visual prompt tuning. When designing our algorithm, we consider the following two questions: (1) how to design informative visual prompts for source-free domain adaptive semantic segmentation and (2) how to learn visual prompts with unlabeled samples for model adaptation. To this end, we propose a universal unsupervised visual prompt tuning (Uni-UVPT) framework as shown in Figure 1. Specifically, we design a novel prompt adapter for generating informative visual prompts and improving the generalization of target features. Besides, an effective adaptive pseudo-label correction strategy with a multiscale consistency loss is proposed for learning visual prompts with massive unlabeled target data and enhancing visual prompts' capacity for spatial perturbations. As the backbone $f_b$ is frozen and only the parameters of the proposed prompt adapter and segmentation head are optimized during training, Uni-UVPT is parameter-efficient for model adaptation. Besides, Uni-UVPT could be directly deployed to various transformer-based architectures without modifying the basic units.

## 2.1  Prompt Adapter

As shown in Figure 1, the proposed lightweight prompt adapter contains a prompt generator and several prompt interactors. The former aims to generate informative prompts that capture multiscale spatial information and task-shared knowledge. The latter is designed to refine prompts with pre-trained knowledge and transform target features to match the pre-trained knowledge in the backbone. To interact with target features via visual prompts, we manually partition the backbone $f_b$ into $N$ (usually $N = 4$) stages, each of which is composed of multiple basic units. Given a target image $x_t$, it is first fed into the embedding layer to obtain patch sequences $F_0^{out}$. $F_0^{out}$ is then regarded as an input of the frozen backbone. Meanwhile, the target image $x_t$ is also fed into the prompt generator to output an initial prompt $C_0$ that contains $L$ dimensional spatial features of multiple resolutions. Then, the initial prompt is flattened and concatenated as an input of the prompt interactor. For stage $i$ of the backbone, its input feature and output feature are respectively denoted as $F_i^{in}$ and $F_i^{out}$. Before stage $i$, the prompt interactor incorporates the output feature $F_{i-1}^{out}$ and the interim prompt $C_{i-1}$ of previous stage to obtain the refined prompts $C_i$ and a suitable input $F_i^{in}$ for the stage $i$. Finally, the output feature $F_N^{out}$ of the last stage is fed into head $f_h$ for segmentation prediction. In the following sections, we introduce the details of the proposed prompt generator and prompt interactor.

**Prompt Generator.** Previous methods [4, 18] have proved that learning prompts brings flexibility to the pre-trained model. However, their prompts like input tokens or pixel-level perturbations are black boxes with limited learning capacity, which cannot reliably explore convincing knowledge beneficial for semantic segmentation. Therefore, we propose to design informative prompts for each image to capture multiscale spatial prior and the task-shared knowledge. To achieve this goal, we utilize a standard convolutional stem $f_{stem}$ borrowed from ResNet [16] to extract multiscale spatial information, because convolution could help transformers better capture the local spatial information [41, 42, 13, 34]. $f_{stem}$ consists of three convolutions, a max-pooling layer, and a stack of stride-2 $3 \times 3$ convolutions to double the channels and reduce the size of feature maps. Several $1 \times 1$ convolutions are applied at the end to project the feature maps to $L$ dimensions. Finally, we obtain a feature pyramid $S = \{s_1, s_2, s_3\}$, which contains $L$-dimensional feature maps with 1/2, 1/4, and 1/8 resolutions of the original image. Besides, following the spirits of previous methods [4, 18], we leverage a group of trainable vectors $Q = \{q_1, q_2, q_3\}$ named level embedding, to learn task-shared knowledge, which contains three $L$-dimensional vectors initialed by the Gaussian function. In this way, the prompt pyramid $\hat{C}_0 = \{c_1, c_2, c_3\}$ of the input image $x_t \in \mathbb{R}^{3 \times H \times W}$ could be obtained by

$$c_i = \kappa\left(s_i, q_i\right), \tag{1}$$

where $\kappa\left(\cdot, \cdot\right)$ could be any feature fusion operations. Here, we first repeat $s_i$ several times so that the vectors can be resized as the same shape of $q_i$, and then perform element-wise addition. Finally, we flatten and concatenate the prompt pyramid $\hat{C}_0$ into prompt tokens $C_0 \in \mathbb{R}^{\left(\frac{HW}{2^2} + \frac{HW}{4^2} + \frac{HW}{8^2}\right) \times L}$ as the input of prompt interactor for prompt refinement and feature adaptation.

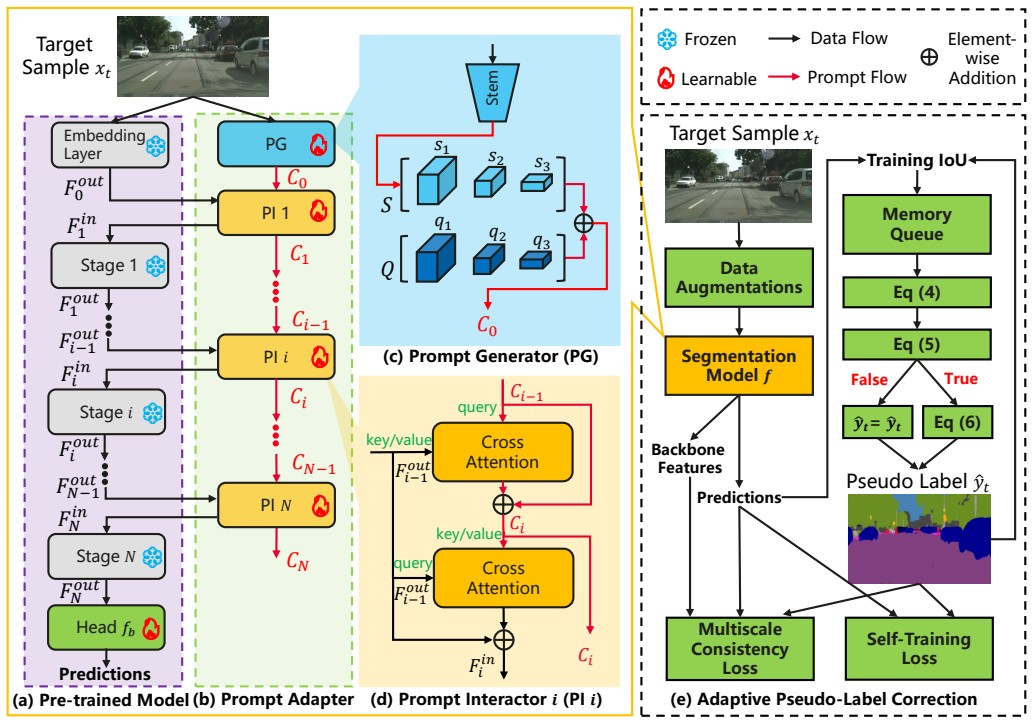

Figure 1: The architecture of Uni-UVPT. (a) The source pre-trained segmentation model, whose encoder layers are divided into $N$ stages. (b) The proposed prompt adapter, which contains two designs, including (c) a prompt generator for encoding rich knowledge into prompts, and (d) a prompt interactor for prompt refinement and feature interaction. (e) The proposed adaptive pseudo-label correction strategy, which rectifies pseudo-labels at suitable moments.

**Prompt Interactor.** To progressively refine prompts and improve feature generalization via the refined prompts, we design a novel prompt interactor based on cross-attention. As shown in Figure 1, for the $i$-th stage of the backbone, we use the interim prompt $C_{i-1}$ of the previous stage as the query, and the output feature $F_{i-1}^{out}$ as the key and value for cross-attention so that the pre-trained knowledge contained in $F_{i-1}^{out}$ are injected into prompts. This process can be formulated as:

$$C_i = C_{i-1} + \text{Attention}(\text{norm}(C_{i-1}), \text{norm}(F_{i-1}^{out})), \qquad (2)$$

where the $\text{norm}(\cdot)$ is LayerNorm [2], and the attention layer $\text{Attention}(\cdot)$ suggests using sparse attention. After that, we utilize the refined prompt $C_i$ to generate adapted input $F_i^{in}$ for the next stage of the backbone. This process can be formulated as:

$$F_i^{in} = F_{i-1}^{out} + \gamma_i \cdot \text{Attention}(\text{norm}(F_{i-1}^{out}), \text{norm}(C_i)), \qquad (3)$$

where sparse attention is adopted to reduce computational costs. Besides, we apply a learnable vector $\gamma_i \in \mathbb{R}^L$ to balance the attention layer's output and the feature $F_{i-1}^{out}$, which is initialized with 0. This initialization strategy ensures that the feature distribution of $F_{i-1}^{out}$ will not be modified drastically due to the injection of the prompt $C_i$, thus making better use of the pre-trained knowledge of backbone.

## 2.2 Adaptive Pseudo-Label Correction

To learn visual prompts with massive unlabeled target data, we propose to utilize high-quality pseudo labels of target samples for self-training. The pipline of the adaptive pseudo-label correction is shown in Figure 1 (e). Previous SFUDA methods generate pseudo labels by learning meaningful cluster structures in the feature space and the quality of the learned cluster structures hinges on the reliability of pseudo labels generated by the source model. Therefore, the pseudo labels are noisy due to the domain shift. Recently, Li et al. [47] have formulated SFUDA as

learning with label noise problem, and observed early-learning phenomenon that deep neural networks tend to first fit the training data with correct pseudo labels during an early-learning phase, before eventually memorizing the instance with incorrect/noisy pseudo labels. Not only in image classification, we also observe a similar phenomenon in semantic segmentation, where the pseudo-label noise is ubiquitous across samples and distributed in a pixel-wise manner. As shown in Figure 2, we analyze the learning curves on the pseudo labels predicted by the source model during the training process. As the model learning is supervised by noisy pseudo labels, the $\text{IoU}_m$ curves for all categories increase substantially as training proceeds. The $\text{IoU}_{el}$ follows a completely different trajectory: it first increases during an early-learning stage where the model learns to correctly segment the incorrectly-labeled pixels, but eventually decreases as memorization occurs.

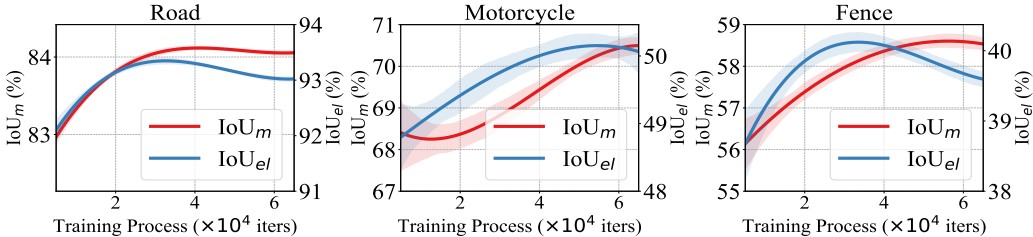

Figure 2: Effect of early learning ($\text{IoU}_{el}$, blue curves) and memorization ($\text{IoU}_m$, red curves) for different categories in Cityscapes dataset. The $\text{IoU}_{el}$ is the IoU between the model output and ground-truth labels. The $\text{IoU}_m$ represents the IoU between the model output and pseudo labels.

To alleviate the memorization of noisy pseudo labels in SFUDA, Li et al. [47] roughly encourage model predictions to stick to labels generated by the frozen source model without any label correction. Differently, we design an adaptive pseudo-label correction strategy based on the early-time training phenomenon, which rectifies target pseudo labels for each category before the model memorizes the noisy pseudo labels. Naturally, it raises a question that how to determine when to correct the noisy pseudo labels. Fortunately, as shown in Figure 2, we observe that the segmentation performance on the training set ($\text{IoU}_m$) improves rapidly during early learning, and then much more slowly during memorization. In other words, the performance deceleration indicates whether the model overfits noisy pseudo labels. To estimate the deceleration, we first fit the following exponential parametric model to the training IoU using the least squares:

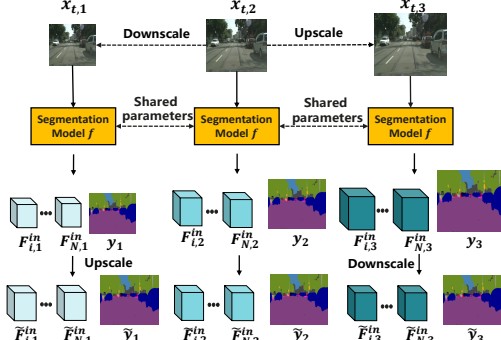

Figure 3: Multiscale consistency loss for rescaled images. The rescaled features of $\tilde{F}_{i,k}^{in}$ of backbone are utilized for feature consistency and the rescaled model predictions $\tilde{y}_k$ for prediction consistency.

$$g(t) = at^3 + bt^2 + ct + d, \qquad (4)$$

where $t$ represents training time, and $a$, $b$, $c$ and $d$ are fitting parameters. However, it is time-consuming to calculate the training IoU on all training samples. We propose to utilize a memory queue with $r$ samples as an alternative, which is updating with batch data. Then, we compute the derivative $g'(t)$, and the pseudo labels for each category are corrected when the relative change of $g'(t)$ is above a certain threshold $\tau$:

$$\frac{|g'(t_0) - g'(t)|}{|g'(t_0)|} > \tau. \qquad (5)$$

where $t_0$ is the start time of the early-learning stage in a correction loop. The image pixels with predictive confidence over $\gamma\%$ per class on the entire target training set are chosen for pseudo-label correction. For all experiments, we set $\tau = 0.9$ and $\gamma = 0.65$. After the pseudo-label correction, the model learning steps into a new early-learning stage. It is noticed that the pseudo-label correction loop will perform several times until achieving the max training iterations.

To form the final pseudo labels, we average the model outputs corresponding to multiple rescaled copies of inputs. Specifically, let $m$ be the number of scaling operations. We set $m = 3$ (downscaling $\times 0.5$, no scaling, and upscaling $\times 2$) in our experiments. For an input $x_t$ rescaled according to these scaling operations in Figure 3, $y_k, 1 \le k \le m$ denotes the corresponding model predictions and $\tilde{y}_k$ is obtained by rescaling $y_k$ as the same size of $x_t$. The corrected pseudo label $\hat{y}_t$ can be obtained by averaging predictions of multiple rescaled $x_t$:

$$\hat{y}_t = \frac{1}{m} \sum_{k=1}^{m} \tilde{y}_k, \tag{6}$$

where only categories satisfying Eq (5) require correction while others remain unchanged.

Once the pseudo labels of all training samples are corrected, the model steps into a new early-learning stage via minimizing a self-training loss $\mathcal{L}_{st}$ shown in Eq (7) and waits for the next correction process.

$$\mathcal{L}_{st} = \mathbb{E}_{x_t \sim \mathcal{D}_t} \left[ -\langle \hat{y}_t, \log f(x_t) \rangle \right] + \mathbb{E}_{x_t \sim \mathcal{D}_t} \left[ -\log f(x_t) \right], \tag{7}$$

where $\langle \cdot, \cdot \rangle$ is dot product, and $f(x_t)$ denotes the model output. The first term is cross-entropy loss based on image and pseudo-label pair $(x_t, \hat{y}_t)$. The second term is predictive entropy that could reduce model uncertainty on target samples during self-training [19].

Besides, we incorporate a multiscale consistency loss $\mathcal{L}_{mc}$ to promote feature consistency and prediction consistency, as shown in Eq (8),

$$\mathcal{L}_{mc} = \alpha \, \mathbb{E}_{x_t \sim \mathcal{D}_t} \underbrace{\left[ \sum_{i=1}^{N} \frac{1}{m} \sum_{k=1}^{m} \| \tilde{F}_{i,k}^{in} - \hat{F}_i^{in} \|_2^2 \right]}_{\text{feature consistency } \mathcal{L}_{fc}} + \beta \, \mathbb{E}_{x_t \sim \mathcal{D}_t} \underbrace{\left[ -\frac{1}{m} \sum_{k=1}^{m} \mathrm{KL} \left( \tilde{y}_k \parallel \hat{y}_t \right) \right]}_{\text{prediction consistency } \mathcal{L}_{pc}}, \tag{8}$$

where $\alpha$ and $\beta$ are balance parameters, and KL denotes the Kullback-Leibler divergence. As shown in Figure 3, $\tilde{F}_{i,k}^{in}, 1 \le k \le m, 1 \le i \le N$ represents the rescaled features of backbone stages and their average is denoted as $\hat{F}_i^{in} = \frac{1}{m} \sum_{k=1}^{m} \tilde{F}_{i,k}^{in}$. The first term is the feature consistency regularization $\mathcal{L}_{fc}$, which constrains the visual prompts to preserve the semantics of features and meanwhile eliminate the spatial perturbations so that the prompt adapter could generate scale-robust prompts. Similarly, the rest prediction consistency term $\mathcal{L}_{pc}$ could encourage the model to produce predictions that are robust to arbitrary semantic-preserving spatial perturbations. The final objective is defined as the sum of the self-training loss and multiscale consistency loss:

$$\mathcal{L} = \mathcal{L}_{st} + \mathcal{L}_{mc}. \tag{9}$$

More details about the training process are introduced in the appendices.

## 3 Experiment

### 3.1 Evaluation Setup

**Datasets and Metrics.** We extensively evaluate the proposed approach on two popular synthetic-to-real benchmarks, *i.e.*, GTA5 $\rightarrow$ Cityscapes and SYNTHIA $\rightarrow$ Cityscapes. GTA5 [37] is a large-scale driving-game dataset containing 24,966 images with 19 classes for pre-training. Synthia dataset [38] is rendered from a virtual city and contains 9,400 synthetic images with 16 classes for pre-training. The realistic dataset Cityscapes [10] collects street view scenes from 50 different cities with 19 classes, including 2,975 training images and 500 validation images. Following previous methods [30, 19], the semantic segmentation performance is evaluated by calculating the Mean Intersection-over-Union (mIoU) over 19 categories in GTA5 $\rightarrow$ Cityscapes, as well as 13 and 16 (with 3 small-scale categories) categories in SYNTHIA $\rightarrow$ Cityscapes.

**Implementation Details.** Our approach is implemented based on the MMSegmentation framework [9] and one training task requires one NVIDIA Tesla V100 GPU. We deploy Swin-B [31] and MiT-B5 [43] as backbones and DAFormer [17] as decode head. During pre-training in the source domain, the backbone is initialized with weights pre-trained with the ImageNet dataset [12]. We utilize AdamW for optimization. Specifically, the learning rate of Swin-B encoder is set as $6 \times 10^{-6}$ and $4 \times 10^{-6}$ for the MiT-B5 encoder, while the learning rates of the segmentation head and prompt adapter are respectively set as ten and five times of backbone. Our Uni-UVPT framework typically needs 40k-80k iterations with a batch size of 1 until convergence. More details could be found in the appendices.

Table 1: Quantitative evaluations on GTA5 → Cityscapes and SYNTHIA → Cityscapes tasks. Different segmentation architectures: F (FCN8s VGG-16), D (DeepLabv2 ResNet-101), S (Swin-B), M (MiT-B5). FB: whether the backbone is frozen. Params (M): number of trainable parameters. **Bold**: the best results based on different source pre-trained models. (+x.x): mIoU gains over the corresponding source pre-trained models where the best are in red. Underline: the state-of-the-art results. The full table with per-class IoUs is available in the appendices.

| Methods | Arch | FB | Params (M) | GTA5 → Cityscapes | SYNTHIA → Cityscapes | |
|---|---|---|---|---|---|---|
| | | | | $\text{mIoU}_{19}(\%)$ | $\text{mIoU}_{16}(\%)$ | $\text{mIoU}_{13}(\%)$ |
| SFDA [30] | F | ✗ | - | 35.8 | - | - |
| GtA [19] | F | ✗ | 134.5 | 45.9 | 41.3 | 48.9 |
| URMA [14] | D | ✗ | 47.4 | 45.1 | 39.6 | 45.0 |
| SRDA [5] | D | ✗ | - | 45.8 | - | - |
| SFUDA [46] | D | ✗ | - | 49.4 | - | 51.9 |
| BDT [20] | D | ✗ | 43.8 | 52.6 | - | 56.7 |
| GtA [19] | D | ✗ | 43.8 | 53.4 | 52.0 | 60.1 |
| Standard Single Source | S | ✗ | 90.7 | 50.5 | 44.6 | 49.8 |
| CPSL [24] | S | ✓ | 3.9 | 51.1 (+0.6) | 46.4 (+1.8) | 52.3 (+2.5) |
| VPT [18] + ELR [47] | S | ✓ | 7.0 | 53.5 (+2.0) | 47.7 (+3.1) | 53.2 (+3.4) |
| *Ours* | S | ✓ | 28.6 | **56.2** (+5.7) | **52.6** (+8.0) | **59.4** (+9.6) |
| Standard Single Source | M | ✗ | 85.2 | 52.5 | 48.6 | 55.0 |
| CPSL [24] | M | ✓ | 3.7 | 52.5 (+0.0) | 50.5 (+1.9) | 57.2 (+2.1) |
| VPT [18] + ELR [47] | M | ✓ | 7.6 | 54.1 (+1.6) | 51.6 (+3.0) | 58.0 (+3.0) |
| *Ours* | M | ✓ | 12.3 | **54.2** (+1.7) | **52.6** (+4.0) | **59.3** (+4.3) |
| Source-GtA [19] | S | ✗ | 110.4 | 52.8 | 48.8 | 55.0 |
| CPSL [24] | S | ✓ | 3.9 | 53.5 (+0.7) | 49.6 (+0.8) | 56.2 (+1.2) |
| VPT [18] + ELR[47] | S | ✓ | 7.0 | 55.1 (+2.3) | 51.6 (+2.8) | 58.2 (+3.2) |
| GtA [19] | S | ✓ | 23.6 | 56.1 (+3.3) | 52.5 (+3.7) | 58.7 (+3.7) |
| *Ours* | S | ✓ | 28.6 | **56.9** (+4.1) | **53.8** (+5.0) | **60.4** (+5.4) |
| Source-GtA [19] | M | ✗ | 103.7 | 53.0 | 50.0 | 56.2 |
| CPSL [24] | M | ✓ | 3.7 | 53.2 (+0.2) | 52.2 (+2.2) | 58.7 (+2.5) |
| VPT [18] + ELR [47] | M | ✓ | 7.6 | 54.4 (+1.4) | 53.0 (+3.0) | 59.5 (+3.3) |
| GtA [19] | M | ✓ | 22.3 | 55.2 (+2.2) | 53.6 (+3.6) | 59.7 (+3.5) |
| *Ours* | M | ✓ | 12.3 | **56.1** (+3.1) | **53.8** (+3.8) | **60.1** (+3.9) |

## 3.2 Comparative Results

**Baselines.** We compare the proposed method with state-of-the-art source-free domain adaptive semantic segmentation methods, namely, SRDA [5], SFUDA [46], SFDA [30], URMA [14], BDT [20], and GtA [19]. We directly cite their results from their original papers. All the above methods deploy CNN backbones, not transformer ones. For a fair comparison, we implement GtA with transformer backbones based on official code [2]. Besides, we utilize representative self-training with pseudo-label methods for unsupervised DA or source-free UDA, i.e., CPSL [24] and ELR [47], and visual prompt tuning method, i.e., VPT [18], to construct several competitors. Our baseline methods, CPSL [24], VPT [18]+ELR [47], GtA [19] are implemented transferring the authors' official releases [3] [4] to MMSegmentation framework [9]. We employ two different methods for source pre-training, i.e., standard single-source and Source-GtA [19]. The former directly utilizes source images and annotations for pre-training. The latter augments source data with five augmentations, and leverages auxiliary heads to learn powerful representations [19].

**Results Analysis.** We compare the proposed approach and the state-of-the-art SFUDA methods on GTA5 → Cityscapes and SYNTHIA → Cityscapes tasks. As shown in Table 1, the proposed method with small amounts of learnable parameters performs better than comparative methods in two transfer tasks. For example, our method with 28.6M (31.5% of the source pre-trained model) trainable parameters increases mIoU by 8.0% (16 classes) and 9.6 % (13 classes) in SYNTHIA → Cityscapes task. The impressive performance demonstrates that the proposed unsupervised visual prompt tuning method could efficiently and effectively adapt large-scale source pre-trained model to the unlabeled target domain. Besides, we could conclude several interesting conclusions. (1)

---

[2]GtA: https://sites.google.com/view/sfdaseg

[3]VPT: https://github.com/KMnP/vpt

[4]CPSL: https://github.com/lslrh/CPSL

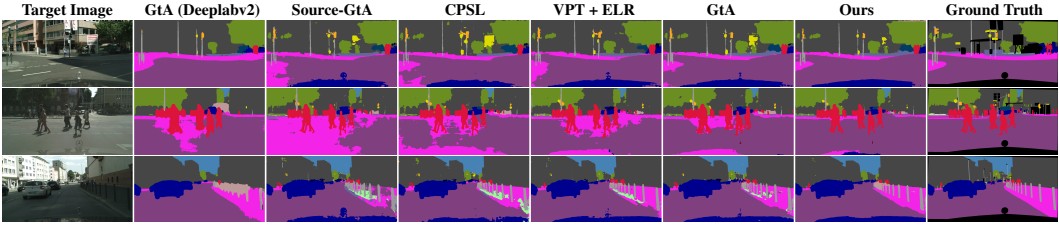

| Target Image | GtA (Deeplabv2) | Source-GtA | CPSL | VPT + ELR | GtA | Ours | Ground Truth |

Figure 4: Qualitative results on GTA5 → Cityscapes.

Table 2: Ablation study on the prompt adapter. PG, PI and LE respectively denote prompt generator, prompt interactor and level embedding. The performance drop is over our complete approach.

| PG | | PI | mIoU (%) |
|---|---|---|---|
| Stem | LE | | |
| Multiscale | ✓ | ✓ | 56.24 (Ours) |
| Multiscale | ✗ | ✓ | 55.58 ↓0.66 |
| Singlescale | ✓ | ✓ | 55.52 ↓0.72 |
| ✗ | ✓ | ✓ | 55.50 ↓0.74 |
| Multiscale | ✓ | PI 1 | 55.34 ↓0.90 |
| ✗ | ✗ | ✗ | 55.07 ↓1.17 |

Table 3: Analysis on pseudo-label strategies.

| Methods | mIoU (%) |
|---|---|
| Ours | 56.24 |
| ELR [47] | 55.60 |
| Ours + offline | 55.47 |

Table 4: Analysis on consistency loss.

| Feature | Prediction | mIoU (%) |
|---|---|---|
| ✓ | ✓ | 56.24 (Ours) |
| ✗ | ✓ | 54.26 |
| ✓ | ✗ | 56.01 |
| ✗ | ✗ | 53.81 |

Compared with SFUDA methods with CNN backbones, SFUDA methods with frozen transformer backbones significantly improve the performance of the target domain, illustrating that it is necessary to adapt the large-scale pre-trained model in domain adaptive semantic segmentation tasks. (2) CPSL underperforms other competitors, illustrating that simply applying pseudo-label correction is not effective in SFUDA. A similar conclusion has been claimed in ELR [47]. (3) VPT + ELR is a strong baseline by combining the visual prompt tuning and SFUDA pseudo-label strategies, which performs worse than our approach in all experimental settings. The possible reasons are as follows: First, visual prompts in VPT are designed as input tokens that have limited capacities to encode informative knowledge for semantic segmentation. Differently, our approach could capture rich information including multiscale spatial information and task-shared knowledge. Second, ELR roughly encourages model predictions to stick to labels generated by the frozen source model without any label correction. We argue that pseudo-label correction is still working but should be performed at suitable moments before the model memorizes noisy pseudo labels, which is supported by our superior performances. More analyses about the pseudo-label strategy refer to the ablation study. As shown in Figure 4, we provide some qualitative results of comparative methods in GTA5 → Cityscapes task. The backbone of none-CNN-based methods is Swin-B, which is pre-trained via Soruce-GtA. Remarkably, our method performs favorably in comparison to other competitors. More visualization results are available in the appendices.

### 3.3 Ablation Study

We have conducted extensive experiments on GTA5 → Cityscapes to study the contributions of each component in the proposed method. Please refer to the appendices for more details.

**Whether can the prompt adapter learn reasonable prompts and facilitate feature generalization?** The prompt adapter consists of a prompt generator (PG) and several prompt interactors (PI). The former can be further divided into stem and level embedding (LE) modules. As shown in Table 2, we have several interesting findings: (1) Without the level embedding, the mIoU drops 0.66%, illustrating that the level embedding could capture task-shared knowledge beneficial for the target domain. (2) To analyze the impact of the stem module, we deploy two variants. One is to replace multiscale feature maps with one feature map. The other is obtained by removing the stem module and repeating level embeddings to the same shape of visual prompts. Compared with single-scale feature map, multiscale feature maps bring 0.72% mIoU improvements. Besides, the mIoU drops 0.74% without the stem module, proving that multiscale spatial knowledge is helpful for prompt learning. (3) To testify whether multiple prompt interactors are necessary, we remove PI $i, 2 \leq i \leq N$

and only preserve PI 1 for transforming the image token sequences. The 0.9% performance drop illustrates that it is necessary to perform prompt refinement and feature interaction between adjacent stages of the backbone. (4) Without the proposed prompt adapter, the performance decreases by 1.17%. The performance drop is smaller than other variants because head tuning also benefits feature adaptation. Despite that, it proves that the prompt adapter could capture informative knowledge for reasonable prompt learning and enhancing the generalization of target features.

**Whether is it necessary to perform pseudo-label correction in source-free UDA?** Li et al. [47] claim that pseudo-label correction strategies perform poorly due to the early-learning phenomenon and even no better than directly supervising the target model with pseudo labels predicted by the frozen source model. We argue that the pseudo-label correction should be conducted at suitable moments before the model memorizes noisy pseudo-labels. As shown in Table 3, compared with ELR [47], the proposed adaptive pseudo label correction strategy improves mIoU by 0.64%, demonstrating that correcting pseudo-label at suitable moments helps to learn more trustable pre-trained knowledge. As for *Ours + offline*, we first train the model to get the training IoU curve for each category and then correct pseudo labels with our strategy only once. The mIoU of *Ours + offline* decreases 0.77%, illustrating that pseudo-label correction should be performed adaptively otherwise the noisy pseudo labels are memorized again after several training epochs.

**Effectiveness of multiscale consistency loss.** The multiscale consistency loss performs feature and prediction consistency of rescaled inputs. As shown in Table 4, without feature consistency, the mIoU drops dramatically as visual prompts can not guarantee that features of different scales of one image are similar. In other words, the feature consistency enhances the robustness of visual prompts to spatial perturbations. Similarly, prediction consistency benefits generating scale-robust predictions. The overall consistency loss increases mIoU by 2.43%.

## 3.4 Further Remark

**Comparative Results of Different Augmentations.** To analyze the influences of other augmented samples, we replace scale augmentation with weather augmentation, e.g., snow and frost. That is, we replace downscale/original/upscale images with snow/original/frost images for unsupervised visual prompt tuning. We summarize the performance of Ours and Ours-weather in Table 5. Ours refers to our approach trained with scale-augmented samples. Ours-weather is our approach trained with weather-augmented samples. Compared with Ours, Ours-weather improves mIoU approximately -1.5% to -0.1% in GTA → Cityscapes task and -0.1 % to +1.6% in SYNTHIA → Cityscapes task. The experimental results illustrate that other augmented samples also work well for unsupervised visual prompt tuning and the SOTA results in SYNTHIA →Cityscapes tasks are updated by weather augmentation.

Table 5: Comparative results of different augmentations on GTA5 → Cityscapes and SYNTHIA → Cityscapes tasks. Different segmentation architectures: S (Swin-B), M (MiT-B5). FB: whether the backbone is frozen. Params (M): number of trainable parameters. (+x.x): mIoU gains over the corresponding source pre-trained models.

| Methods | Arch | FB | Params (M) | GTA5 → Cityscapes | SYNTHIA → Cityscapes |
|---|---|---|---|---|---|
| | | | | $mIoU_{19}(\%)$ | $mIoU_{16}(\%)$ |
| Ours | S | ✓ | 28.6 | 56.2 (+5.7) | 52.6 (+8.0) |
| Ours-weather | S | ✓ | 28.6 | 54.7 (+4.2) | 52.9 (+8.3) |
| Ours | M | ✓ | 12.3 | 54.2 (+1.7) | 52.6 (+4.0) |
| Ours-weather | M | ✓ | 12.3 | 54.1 (+1.6) | 53.0 (+4.4) |
| Ours | S | ✓ | 28.6 | 56.9 (+4.1) | 53.8 (+5.0) |
| Ours-weather | S | ✓ | 28.6 | 54.1 (+1.6) | 53.0 (+4.4) |
| Ours | M | ✓ | 12.3 | 56.1 (+3.1) | 53.8 (+3.8) |
| Ours-weather | M | ✓ | 12.3 | 55.2 (+2.2) | 54.5 (+4.5) |

**Visualization Results of Adaptive Pseudo-Label Correction Strategy.** To emphasize the effectiveness of the proposed adaptive pseudo-label correction strategy, we draw the $IoU_{el}$ and $IoU_m$ curves for each category as shown in Figure 5. We observe that the $IoU_{el}$ curves of all categories increase until the model converges and the phenomenon of $IoU_{el}$ decreasing due to the noise memorization are disappeared. The above interesting observations demonstrate that the proposed adaptive pseudo-label

correction strategy could effectively alleviate the negative effect caused by pseudo-label noises. It also proves that pseudo-label correction works in source-free unsupervised domain adaptation when it is performed at suitable moments.

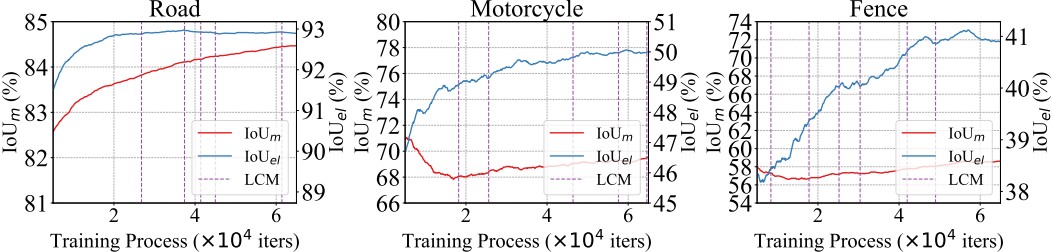

Figure 5: Analysis on early learning ($IoU_{el}$, blue curves) and memorization ($IoU_m$, red curves) with our proposed adaptive pseudo-label correction strategy for different categories in Cityscapes dataset. The LCM (Label Correction Moment) denotes the moment when the algorithm performs pseudo-label correction.

## 4    Related work

**Source-Free Unsupervised Domain Adaptation for Semantic Segmentation.** Source-free unsupervised domain adaptation has attracted much attention when source data is not available during model adaptation. Most SFUDA methods [21, 23, 26] are proposed for classification task, which is suboptimal for cross-domain semantic segmentation. Only a few methods [30, 19, 20, 46] focus on source-free domain adaptive semantic segmentation. For example, SFDA [30] not only recovers and preserves the source domain knowledge via knowledge transfer, but also distills valuable information from the target domain for self-supervised learning. Kundu et al. [19] design a multi-head framework to extract reliable target pseudo labels for self-training. Kundu et al. [20] further introduce an intermediate mixup domain between the original and the realizable generic domain, which could enhance the trade-off between discriminability and transferability. Ye et al. [46] improve the target domain performance by reconstructing the source domain data and performing distribution-aware adversarial learning and self-training. However, current source-free methods adopt fine-tuning for model adaptation, which is inefficient for large-scale transformer-based architectures.

**Visual Prompt Tuning.** Recently, visual prompt tuning has been introduced into the vision area [3, 32, 40] to generalize the large-scale pre-trained model to downstream tasks. Previous methods design visual prompts from the view of token-level, pixel-level and module-level. Token-level prompt tuning methods [18, 15, 33, 40] govern learnable visual tokens to the embedding layer or several encoder layers. Pixel-level prompt tuning methods [4, 6, 36] introduce learnable pixel perturbations on the original images. However, token-level or pixel-level prompts are black boxes, which cannot capture convincing knowledge beneficial for downstream tasks. Module-level prompt tuning methods [49, 48, 7] design auxiliary layers or blocks for the pre-trained model. Despite learning rich knowledge, they are specifically designed for the pre-trained model by modifying its basic units, which can not be applied to various architectures. As far as we know, only [15] has applied visual prompts for online unlabeled adaption in image classification, and none of the work has explored unsupervised visual prompt tuning for source-free domain adaptive semantic segmentation.

## 5    Conclusion and Discussion

In this paper, we propose a universal unsupervised visual prompt tuning (Uni-UVPT) framework, which adapts a large-scale pre-trained source model for source-free domain adaptive semantic segmentation. A novel prompt adapter is proposed to progressively encode informative knowledge into prompts and enable target features to match the pre-trained model. Cooperatively, an adaptive pseudo-label correction strategy with a multiscale consistency loss is proposed to correct pseudo-label at suitable moments and enhance visual prompts' spatial robustness. Extensive experiments prove that our approach could effectively and efficiently perform large-model unsupervised knowledge transfer. Our approach also has its limitations, for example, further reducing the trainable parameters by delicately designing prompt modules and exploring more powerful learning constraints for visual prompts, *e.g.*, pixel-wise contrastive loss. We hope this paper could inspire researchers to further investigate unsupervised visual prompt tuning for other vision applications.

## Acknowledgments

We gratefully acknowledge the support of MindSpore, CANN (Compute Architecture for Neural Networks) and Ascend AI Processor used for this research.

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
