# Appendix

## A  Further Implementation Details

### A.1  Network Architectures

We adopt Daformer [17] with Swin-B or MiT-B5 backbone as the base semantic segmentation architecture. For the segmentation head, we utilize the same head as Daformer [17]. The stem module contains one fully-convolutional layers with kernel $3\times3$ and stride of 2, two fully-convolutional layers with kernel $3\times3$ and stride of 1, two fully-convolutional layers with kernel $3\times3$ and stride of 2, and another three fully-convolutional layers with kernel $1\times1$ and stride of 1 to adjust channels of different feature maps. Level embedding module is defined as metrics with shape $3\times$dims. The prompt Interactor module contains three fully-convolutional layers with kernel $3\times3$ and stride of 2 to adjust feature dimensions. The prompt dimensions of different stages keep the same as the backbone, *i.e.*, [128, 256, 512, 1024] for swin-B and [64, 128, 320, 512] for MiT-B5.

### A.2  Training and Testing Details

In the main manuscript, we have provided the most important part of training and testing details. Here, we add other details about the training procedure, baseline implementations, etc. The training procedure of our universal unsupervised visual prompt tuning is summarized in Algorithm 1. Both source and target images are randomly cropped and resized along with randomized horizontal flipping and photometric jittering. Specifically, We resize the images in GTA5 and Cityscapes datasets to $1024\times512$ and resize the images in SYNTHIA dataset to $1280\times760$. Then, all images are randomly cropped to $512\times512$ for pre-training and self-training following [45]. For evaluation, we use the Cityscapes training dataset for training and the standard validation set for testing [45]. We conduct our experiments using PyTorch v1.10.2, CUDA v10.2, and CuDNN v7.6.5, and all experiments are done on a single NVIDIA V100 GPU.

### A.3  Variant Implementation in Ablation Study

In this section, we introduce the implementation details about model variants in ablation study (Section 3.3) of the main manuscript.

**Prompt Adapter.** To analyze the effectiveness of the proposed prompt adapter, we study five model variants. (1) w/o Level Embedding. We remove the level embeddings $Q$ and directly regard the output of the stem $\hat{C}_0$ as visual prompts. (2) w/o Stem. We first remove the stem module and then repeat level embeddings to the same shape as visual prompts. The repeated level embeddings are treated as visual prompts. (3) Singlescale Stem. Only the feature map with medium size (1/4 resolution, $c_2$ in $\hat{C}_0$) is applied for prompt generation. (4) Single Prompt Interactor. We only preserve the prompt interactor in the first stage of the backbone and prompt interactors of the remaining stages are removed. (5) w/o Prompt Adapter. We remove the proposed prompt adapter and preserve the backbone for feature extractions, thus only parameters of the segmentation head are optimized during self-training with pseudo labels.

**Adaptive Pseudo Label Correction.** To analyze the impact of the proposed adaptive pseudo-label correction strategy, we conduct experiments on two model variants. (1) ELR. The model is learned with pseudo-labels predicted by the frozen source model without any pseudo-label correction strategies. (2) Ours + offline. We first train the model with pseudo labels predicted by the frozen source model to get the training IoU curve for each category, and then correct pseudo labels with our strategy only once.

**Multiscale Consistency Loss.** To analyze the effect of feature consistency $\mathcal{L}_{fc}$ and prediction consistency $\mathcal{L}_{pc}$ in multiscale consistency loss $\mathcal{L}_{mc}$: we conduct three objectives. (1) w/o feature consistency. The feature consistency term is removed from the final objective which is defined as $\mathcal{L} = \mathcal{L}_{st} + \mathcal{L}_{pc}$. (2) w/o prediction consistency. We remove the prediction consistency term and the learning objective is described as $\mathcal{L} = \mathcal{L}_{st} + \mathcal{L}_{fc}$. (3) w/o multiscale consistency loss. The final loss function only contains self-training loss, *i.e.*, $\mathcal{L} = \mathcal{L}_{st}$.

**Algorithm 1** The overall training procedure of universal unsupervised visual prompt tuning.

---

**Input:** Uni-UVPT model $f$ with trainable parameters $\theta$ (The backbone is frozen.), memory queue $\mathcal{M}$ with length $r$, target data with pseudo labels predicted by source model $f_s$: $(x_t, \hat{y}_t)$, label set $\mathcal{C}$;

**Define:** Fitting interval $\delta$, relative change threshold $\tau$, predictive confidence threshold $\gamma$, multiscale consistency loss weights: $\alpha$ and $\beta$, total training iterations $T$;

```
1:  for x_t ∈ D_t, ŷ_t = f_s(x_t) do
2:      while length of M < r do
3:          M = M ∪ IoU(f(x_t), ŷ_t);
4:      end while
5:      t_0 = 0;
6:      repeat
7:          for each category c in C do
8:              if t mod δ = 0 and t − t_0 ≥ one epoch then
9:                  Calculate total IoU of M and store its value;
10:                 Fit IoU with g(t) = at³ + bt² + ct + d;
11:                 if |g'(t_0)−g'(t)|/|g'(t_0)| > τ then
12:                     Correct the pixel-level pseudo label ŷ of category c with predictive confidence
13:                     over γ %;
14:                     t_0 = t;
15:                 end if
16:             end if
17:             delete M(0);
18:             M = M ∪ IoU(f(x_t), ŷ);
19:             Update θ by minimizing L = L_st + L_mc;
20:         end for
21:     until t ≥ T
22: end for
```

**Output:** model parameters $\theta$;

---

# B   Additional Experimental Results

## B.1   Hyperparameter Analysis.

**Pseudo-label Correction Margin $\tau$ and $\gamma$.** As shown in Figure 6(a), we explore two important hyperparameters, $\tau$ and $\gamma$. $\tau$ decides when to perform pseudo-label correction for each category. $\gamma$ controls which pixel's pseudo label should be corrected. For smaller $\tau$, we observe a small performance drop, illustrating that correcting the pseudo labels too early ($\tau < 0.9$) leads to suboptimal performance. Therefore, a relatively high $\tau$ is preferable, so we set $\tau = 0.9$ for all experiments. Besides, if $\gamma$ is smaller than 0.65, the pseudo-label correction brings too much noise, leading to significant performance degradation. Thus, We set $\gamma = 0.65$ for all experiments.

**Memory Queue Length $r$ and Evaluation Interval $\delta$.** We analyze the influence of the maximum length $r$ of the memory queue and the curve updating interval $\delta$. As shown in Figure 6(b), we find that higher $r$ leads to better performance but brings more computing cost. To balance the computing efficiency and performance, we set $r = 1000$ in our experiments. Meanwhile, we find that the model achieves the best performance when setting $\delta = 200$.

**Loss Weight $\alpha$ and $\beta$.** $\alpha$ is the weight of feature consistency $\mathcal{L}_{fc}$ while $\beta$ represents the weight of prediction consistency $\mathcal{L}_{pc}$. The mIoU performances under different values of $\alpha$ and $\beta$ are shown in Figure 6(c). For $\alpha$, we observe that too small weight ($\alpha < 1$) leads to a significant performance drop, illustrating that feature consistency plays an important role in helping prompt adapter learn scale-robust representations. As for $\beta$, too large weight($\beta > 10^{-3}$) leads to suboptimal mIoU, which may enhance the negative effect of noisy pseudo labels via self-training loss. Thus, we set $\alpha = 1$ and $\beta = 10^{-3}$ in our experiments.

**Learning Rate of the Prompt Adapter.** As shown in Figure 6(d), when the learning rate of the prompt adapter is five times of the head, the model obtains the best performance.

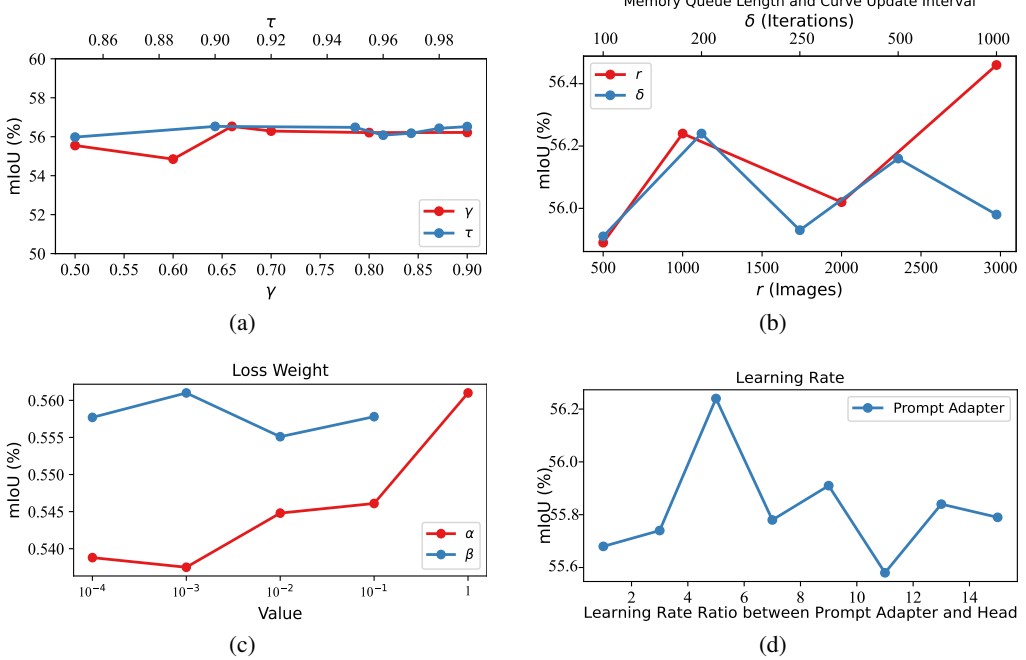

Figure 6: Hyperparameter Analysis: (a) $\tau$ and $\gamma$; (b) $r$ and $\delta$; (c) $\alpha$ and $\beta$; (d) Learning rate of prompt adapter.

## B.2 Quantitative analysis

To quantitatively demonstrate the effectiveness of our approach, we have provided mIoU performance in the main manuscript (see Section 3.2). Here, we provide class-wise IoU results on the GTA5 → Cityscapes and SYNTHIA → Cityscapes benchmarks for semantic segmentation in Table 6 and Table 7. We obtain state-of-the-art performance without fine-tuning the parameters of the backbone. Besides, as shown in Table 8 and Table 9, we conduct comparative results via fine-tuning all parameters including the backbone. Our approach could further improve model generation for the target domain and outperforms other state-of-the-art competitors.

## B.3 Qualitative analysis

We provide extended qualitative results of our proposed approach with two backbones (Swin-B and MiT-B5) and two different pre-training methods (Standard Single Source and Source-GtA) on GTA5 → Cityscapes and SYNTHIA → Cityscapes tasks. Visualization results on eight different settings are shown in Figure 7–14.

Table 6: Quantitative evaluations on GTA5 → Cityscapes. Different segmentation architectures: F (FCN8s VGG-16), D (DeepLabv2 ResNet-101), S (Swin-B), M (MiT-B5). FB: whether the backbone is frozen. Params (M): number of trainable parameters. **Bold**: the best results based on different source pre-trained models. Underline: the state-of-the-art results. SSS: Standard Single Source.

| Method | Arch | FB | Params (M) | $mIOU_{19}$(%) | road | sidewalk | building | wall* | fence* | pole* | light | sign | veg. | terrain | sky | person | rider | car | truck | bus | train | motor | bike |
|---|---|---|---|---|---|---|---|---|---|---|---|---|---|---|---|---|---|---|---|---|---|---|---|
| SFDA | F | ✗ | - | 35.8 | 81.8 | 35.4 | 82.3 | 21.6 | 20.2 | 25.3 | 17.8 | 4.7 | 80.7 | 24.6 | 80.4 | 50.5 | 9.2 | 78.4 | 26.3 | 19.8 | 11.1 | 6.7 | 4.3 |
| GtA | F | ✗ | 134.5 | 45.9 | 92.9 | 56.9 | 82.5 | 20.4 | 6.0 | 30.8 | 34.7 | 33.2 | 84.6 | 17.0 | 88.9 | 62.3 | 30.7 | 85.1 | 15.3 | 40.6 | 10.2 | 30.1 | 50.4 |
| URMA | A | ✗ | 47.4 | 45.1 | 92.3 | 55.2 | 81.6 | 30.8 | 18.8 | 37.1 | 17.7 | 12.1 | 84.2 | 35.9 | 83.8 | 57.7 | 24.1 | 81.7 | 27.5 | 44.3 | 6.9 | 24.1 | 40.4 |
| SRDA | A | ✗ | - | 45.8 | 90.5 | 47.1 | 82.8 | 32.8 | 28.0 | 29.9 | 35.9 | 34.8 | 83.3 | 39.7 | 76.1 | 57.3 | 23.6 | 79.5 | 30.7 | 40.2 | 0.0 | 26.6 | 30.9 |
| SFUDA | A | ✗ | - | 49.4 | - | - | - | - | - | - | - | - | - | - | - | - | - | - | - | - | - | - | - |
| BDT | A | ✗ | 43.8 | 52.6 | - | - | - | - | - | - | - | - | - | - | - | - | - | - | - | - | - | - | - |
| GtA | A | ✗ | 43.8 | 53.4 | 91.7 | 53.4 | 86.1 | 37.6 | 32.1 | 37.4 | 38.2 | 35.6 | 86.7 | 48.5 | 89.9 | 62.6 | 34.3 | 87.2 | 51.0 | 50.8 | 4.2 | 42.7 | 53.9 |
| SSS | S | ✗ | 90.7 | 50.5 | 80.8 | 29.5 | 85.1 | 36.8 | 31.0 | 36.2 | 46.1 | 35.2 | 87.2 | 41.4 | 86.8 | 63.2 | 26.8 | 85.6 | 43.3 | 55.3 | 17.4 | 40.5 | 32.2 |
| CPSL | S | ✓ | 3.9 | 51.1 | 76.0 | 26.9 | 84.7 | 30.6 | 26.7 | 30.1 | 43.8 | 45.9 | 86.8 | 41.8 | 88.1 | 63.7 | 27.1 | 84.8 | 47.6 | 53.0 | 23.3 | 44.4 | 44.8 |
| VPT+ELR | S | ✓ | 7.0 | 53.5 | 85.7 | 35.2 | 85.9 | 40.3 | 29.5 | 38.0 | 46.9 | 37.3 | 88.0 | 45.2 | 90.7 | 64.0 | 30.6 | 86.7 | 45.1 | 59.9 | 28.2 | 39.8 | 39.9 |
| Ours | S | ✓ | 28.6 | **56.2** | 93.2 | 54.0 | 87.4 | 41.9 | 34.2 | 41.6 | 46.5 | 43.0 | 88.6 | 44.7 | 90.9 | 65.8 | 27.9 | 89.0 | 47.0 | 61.2 | 25.8 | 40.3 | 45.4 |
| SSS | M | ✗ | 85.2 | 52.5 | 85.3 | 23.3 | 86.2 | 35.3 | 35.2 | 38.2 | 45.4 | 27.4 | 87.6 | 43.2 | 89.0 | 65.8 | 31.3 | 89.6 | 49.9 | 59.8 | 38.7 | 38.1 | 28.5 |
| CPSL | M | ✓ | 3.7 | 52.5 | 85.9 | 21.5 | 85.8 | 31.9 | 36.9 | 32.8 | 44.1 | 27.1 | 87.3 | 46.8 | 88.3 | 63.9 | 31.0 | 88.5 | 53.6 | 60.4 | 41.9 | 38.1 | 32.1 |
| VPT+ELR | M | ✓ | 7.6 | 54.1 | 85.8 | 22.5 | 86.8 | 33.8 | 38.1 | 39.8 | 48.8 | 27.0 | 88.5 | 45.7 | 89.7 | 67.0 | 32.9 | 90.0 | 54.5 | 60.0 | 42.6 | 38.5 | 36.0 |
| Ours | M | ✓ | 12.3 | **54.2** | 89.4 | 22.9 | 87.3 | 38.9 | 34.3 | 41.0 | 45.8 | 30.3 | 88.8 | 44.2 | 90.1 | 67.0 | 32.4 | 90.1 | 52.9 | 60.4 | 37.1 | 37.7 | 38.6 |
| Source-GtA | S | ✗ | 110.4 | 52.8 | 87.8 | 40.9 | 86.2 | 44.3 | 31.9 | 38.9 | 43.6 | 32.7 | 87.8 | 47.6 | 84.4 | 64.1 | 29.7 | 88.5 | 45.8 | 56.7 | 19.8 | 36.1 | 36.6 |
| CPSL | S | ✓ | 3.9 | 53.5 | 89.1 | 44.2 | 85.5 | 32.1 | 35.1 | 34.7 | 42.3 | 40.1 | 86.8 | 46.7 | 85.1 | 63.2 | 31.4 | 87.0 | 47.5 | 60.2 | 29.9 | 34.1 | 41.5 |
| VPT+ELR | S | ✓ | 7.0 | 55.1 | 90.5 | 47.1 | 86.8 | 40.2 | 32.0 | 40.4 | 44.8 | 32.8 | 88.3 | 50.2 | 86.5 | 65.2 | 33.5 | 89.3 | 49.1 | 57.1 | 33.5 | 38.5 | 41.7 |
| GtA | S | ✓ | 23.6 | 56.1 | 90.0 | 45.4 | 87.7 | 45.3 | 37.0 | 42.0 | 46.3 | 37.8 | 88.6 | 49.8 | 88.0 | 65.9 | 34.3 | 88.3 | 48.2 | 60.0 | 29.1 | 33.7 | 48.1 |
| Ours | S | ✓ | 28.6 | **56.9** | 93.4 | 53.9 | 87.2 | 43.5 | 35.5 | 39.2 | 43.9 | 34.6 | 87.8 | 50.3 | 85.8 | 65.4 | 33.4 | 88.5 | 53.7 | 61.9 | 32.3 | 39.0 | 51.3 |
| Source-GtA | M | ✗ | 103.7 | 53.0 | 84.4 | 32.1 | 86.7 | 48.0 | 33.6 | 39.5 | 46.5 | 31.6 | 87.7 | 48.3 | 86.5 | 64.7 | 29.8 | 89.2 | 51.7 | 55.2 | 35.8 | 32.7 | 23.8 |
| CPSL | M | ✓ | 3.7 | 53.2 | 86.3 | 35.3 | 85.9 | 41.5 | 34.3 | 34.3 | 44.8 | 30.9 | 87.2 | 50.2 | 86.6 | 62.8 | 26.8 | 88.4 | 57.3 | 58.4 | 38.3 | 32.8 | 27.9 |
| VPT+ELR | M | ✓ | 7.6 | 54.4 | 85.5 | 31.7 | 87.0 | 48.7 | 32.8 | 40.2 | 47.6 | 31.2 | 88.1 | 48.2 | 87.7 | 65.6 | 30.7 | 90.0 | 59.3 | 59.6 | 31.3 | 34.9 | 32.7 |
| GtA | M | ✓ | 22.3 | 55.2 | 86.3 | 33.0 | 87.4 | 43.3 | 35.6 | 42.4 | 49.0 | 33.5 | 88.6 | 49.6 | 88.5 | 66.3 | 31.9 | 89.0 | 53.6 | 57.1 | 39.1 | 33.6 | 40.7 |
| Ours | M | ✓ | 12.3 | **56.1** | 87.4 | 35.6 | 87.8 | 48.5 | 36.7 | 41.1 | 47.5 | 38.4 | 88.3 | 49.4 | 88.0 | 66.4 | 31.1 | 89.7 | 54.6 | 58.9 | 35.4 | 35.7 | 45.9 |

Table 7: Quantitative evaluations on SYNTHIA → Cityscapes. Different segmentation architectures: F (FCN8s VGG-16), D (DeepLabv2 ResNet-101), S (Swin-B), M (MiT-B5). FB: whether the backbone is frozen. Params (M): number of trainable parameters. **Bold**: the best results based on different source pre-trained models. Underline: the state-of-the-art results. SSS: Standard Single Source.

| Method | Arch | FB | Params (M) | $mIOU_{16}$(%) | $mIoU_{13}$(%) | road | sidewalk | building | wall* | fence* | pole* | light | sign | veg. | sky | person | rider | car | bus | motor | bike |
|---|---|---|---|---|---|---|---|---|---|---|---|---|---|---|---|---|---|---|---|---|---|---|
| GtA | F | ✗ | 134.5 | 41.3 | 48.9 | 89.9 | 48.8 | 80.9 | 2.9 | 2.5 | 28.1 | 19.5 | 26.2 | 83.7 | 84.9 | 57.4 | 17.8 | 75.6 | 28.9 | 4.3 | 17.2 |
| URMA | A | ✗ | 47.4 | 39.6 | 45.0 | 59.3 | 24.6 | 77.0 | 14.0 | 1.8 | 31.5 | 18.3 | 32.0 | 83.1 | 80.4 | 46.3 | 17.8 | 76.7 | 17.0 | 18.5 | 34.6 |
| SFUDA | A | ✗ | - | - | 51.9 | - | - | - | - | - | - | - | - | - | - | - | - | - | - | - | - |
| BDT | A | ✗ | 43.8 | - | 56.7 | - | - | - | - | - | - | - | - | - | - | - | - | - | - | - | - |
| GtA | A | ✗ | 43.8 | 52.0 | 60.1 | 90.5 | 50.0 | 81.6 | 13.3 | 2.8 | 34.7 | 25.7 | 33.1 | 83.8 | 89.2 | 66.0 | 34.9 | 85.3 | 53.4 | 46.1 | 46.6 |
| SSS | S | ✗ | 90.7 | 44.6 | 49.8 | 71.8 | 31.7 | 77.5 | 26.5 | 0.8 | 37.8 | 37.0 | 29.1 | 79.9 | 82.7 | 59.7 | 23.6 | 66.9 | 47.0 | 22.1 | 19.1 |
| CPSL | S | ✓ | 1.9 | 46.4 | 52.3 | 72.2 | 32.0 | 79.9 | 29.5 | 1.9 | 31.1 | 40.9 | 30.8 | 77.9 | 72.3 | 63.5 | 31.5 | 64.8 | 43.5 | 35.4 | 35.6 |
| VPT+ELR | S | ✓ | 7.0 | 47.7 | 53.2 | 76.8 | 35.3 | 79.6 | 30.9 | 1.2 | 39.4 | 40.8 | 32.4 | 81.8 | 85.1 | 61.8 | 25.2 | 71.3 | 49.8 | 26.3 | 25.7 |
| Ours | S | ✓ | 28.6 | **52.6** | **59.4** | 88.1 | 46.1 | 84.4 | 28.4 | 1.5 | 39.2 | 41.8 | 35.6 | 85.2 | 86.0 | 63.6 | 26.9 | 84.9 | 50.7 | 37.0 | 42.3 |
| SSS | M | ✗ | 85.2 | 48.6 | 55.0 | 84.9 | 36.0 | 81.5 | 23.5 | 2.1 | 37.9 | 37.8 | 34.0 | 79.7 | 83.2 | 63.5 | 29.2 | 77.5 | 46.0 | 34.8 | 26.8 |
| CPSL | M | ✓ | 3.7 | 50.5 | 57.2 | 86.2 | 38.1 | 81.9 | 27.8 | 1.2 | 35.6 | 38.4 | 35.8 | 81.5 | 84.9 | 63.9 | 30.5 | 79.6 | 48.5 | 41.5 | 32.5 |
| VPT+ELR | M | ✓ | 7.6 | 51.6 | 58.0 | 85.4 | 37.7 | 82.6 | 28.7 | 3.3 | 38.8 | 41.1 | 37.8 | 82.3 | 87.0 | 66.2 | 34.2 | 76.4 | 48.2 | 38.8 | 36.5 |
| Ours | M | ✓ | 12.3 | **52.6** | **59.3** | 87.5 | 40.5 | 83.3 | 28.7 | 2.2 | 39.4 | 40.1 | 37.9 | 83.3 | 88.8 | 62.3 | 30.0 | 86.0 | 52.3 | 37.6 | 41.6 |
| Source-GtA | S | ✗ | 110.4 | 48.8 | 55.0 | 85.3 | 37.0 | 81.0 | 25.4 | 1.8 | 39.1 | 38.5 | 30.3 | 78.5 | 79.6 | 61.1 | 30.5 | 82.7 | 42.4 | 30.9 | 37.3 |
| CPSL | S | ✓ | 3.9 | 49.6 | 56.2 | 81.7 | 36.8 | 82.3 | 33.8 | 3.3 | 25.8 | 41.0 | 37.0 | 82.1 | 80.2 | 55.0 | 31.6 | 80.3 | 36.3 | 36.8 | 49.9 |
| VPT+ELR | S | ✓ | 7.0 | 51.6 | 58.2 | 86.6 | 39.7 | 82.1 | 26.4 | 2.1 | 40.4 | 42.8 | 31.7 | 80.3 | 83.1 | 63.3 | 34.5 | 85.2 | 47.4 | 36.8 | 43.6 |
| GtA | S | ✓ | 23.6 | 52.5 | 58.7 | 86.9 | 38.8 | 82.8 | 32.3 | 3.4 | 40.2 | 41.9 | 33.5 | 79.8 | 82.0 | 66.5 | 32.9 | 85.5 | 46.7 | 35.7 | 50.6 |
| Ours | S | ✓ | 28.6 | **53.8** | **60.4** | 88.3 | 43.6 | 84.4 | 32.6 | 2.8 | 40.3 | 43.4 | 29.6 | 83.5 | 87.5 | 67.5 | 34.9 | 85.6 | 49.8 | 34.0 | 53.8 |
| Source-GtA | M | ✗ | 103.7 | 50.0 | 56.2 | 84.9 | 42.3 | 82.0 | 29.3 | 2.1 | 38.1 | 35.5 | 34.4 | 82.0 | 86.5 | 61.7 | 30.2 | 81.6 | 44.4 | 32.8 | 32.3 |
| CPSL | M | ✓ | 3.7 | 52.2 | 58.7 | 86.4 | 44.0 | 83.0 | 34.5 | 2.0 | 35.3 | 38.3 | 35.6 | 83.8 | 87.3 | 62.3 | 31.5 | 85.3 | 47.2 | 39.3 | 39.7 |
| VPT+ELR | M | ✓ | 7.6 | 53.0 | 59.5 | 88.6 | 46.4 | 82.2 | 34.4 | 2.0 | 38.4 | 38.2 | 34.8 | 82.9 | 87.9 | 65.0 | 33.7 | 86.1 | 47.6 | 41.0 | 39.2 |
| GtA | M | ✓ | 22.3 | 53.6 | 59.7 | 87.9 | 46.3 | 83.9 | 35.2 | 3.8 | 41.4 | 35.0 | 35.4 | 83.9 | 89.6 | 68.0 | 33.0 | 85.0 | 44.7 | 37.2 | 46.5 |
| Ours | M | ✓ | 12.3 | **53.8** | **60.1** | 88.6 | 47.8 | 84.0 | 36.8 | 3.0 | 39.8 | 37.3 | 35.4 | 83.9 | 87.2 | 66.2 | 31.3 | 85.0 | 50.6 | 39.1 | 45.0 |

Table 8: Quantitative evaluations without freezing backbone on GTA5 → Cityscapes. Different segmentation architectures: S (Swin-B), M (MiT-B5). FB: whether the backbone is frozen. Params (M): number of trainable parameters. **Bold**: the best results based on different source pre-trained models. Underline: the state-of-the-art results. SSS: Standard Single Source.

| Method | Arch | FB | Params (M) | mIOU$_{19}$(%) | road | sidewalk | building | wall* | fence* | pole* | light | sign | veg. | terrain | sky | person | rider | car | truck | bus | train | motor | bike |
|---|---|---|---|---|---|---|---|---|---|---|---|---|---|---|---|---|---|---|---|---|---|---|---|
| SSS | S | ✗ | 90.7 | 50.5 | 80.8 | 29.5 | 85.1 | 36.8 | 31.0 | 36.2 | 46.1 | 35.2 | 87.2 | 41.4 | 86.8 | 63.2 | 26.8 | 85.6 | 43.3 | 55.3 | 17.4 | 40.5 | 32.2 |
| CPSL | S | ✗ | 90.7 | 53.0 | 82.5 | 31.1 | 86.2 | 34.3 | 31.2 | 37.9 | 45.5 | 46.6 | 87.1 | 41.6 | 87.7 | 65.7 | 29.7 | 88.1 | 48.1 | 53.7 | 24.6 | 33.6 | 51.4 |
| VPT+ELR | S | ✗ | 93.8 | 54.8 | 84.0 | 35.1 | 86.7 | 43.2 | 35.1 | 38.3 | 49.2 | 38.7 | 88.4 | 45.4 | 90.2 | 65.8 | 31.1 | 88.2 | 47.7 | 63.6 | 34.5 | 42.1 | 34.7 |
| Ours | S | ✗ | 115.4 | **57.0** | 93.6 | 56.0 | 87.6 | 45.5 | 37.1 | 41.2 | 48.1 | 36.8 | 88.8 | 45.4 | 90.3 | 67.1 | 32.1 | 89.0 | 53.5 | 63.9 | 27.0 | 38.8 | 40.6 |
| SSS | M | ✗ | 85.2 | 52.5 | 85.3 | 23.3 | 86.2 | 35.3 | 35.2 | 38.2 | 45.4 | 27.4 | 87.6 | 43.2 | 89.0 | 65.8 | 31.3 | 89.6 | 49.9 | 59.8 | 38.7 | 38.1 | 28.5 |
| CPSL | M | ✗ | 85.2 | 52.9 | 85.6 | 23.6 | 85.9 | 32.7 | 37.8 | 32.7 | 44.0 | 26.7 | 87.5 | 47.7 | 88.3 | 63.8 | 30.5 | 89.0 | 56.5 | 60.3 | 36.7 | 40.9 | 34.4 |
| VPT+ELR | M | ✗ | 89.1 | 54.4 | 86.8 | 22.9 | 86.8 | 28.8 | 36.2 | 40.4 | 47.8 | 29.5 | 88.2 | 46.2 | 90.1 | 66.5 | 32.9 | 90.3 | 52.2 | 64.8 | 49.4 | 41.5 | 33.7 |
| Ours | M | ✗ | 93.7 | **55.9** | 89.9 | 26.8 | 87.7 | 44.8 | 36.3 | 39.7 | 48.1 | 30.7 | 88.5 | 45.6 | 90.1 | 66.8 | 32.4 | 90.9 | 60.5 | 65.9 | 41.8 | 37.1 | 38.6 |
| Source-GtA | S | ✗ | 110.4 | 52.8 | 87.8 | 40.9 | 86.2 | 44.3 | 31.9 | 38.9 | 43.6 | 32.7 | 87.8 | 47.6 | 84.4 | 64.1 | 29.7 | 88.5 | 45.8 | 56.7 | 19.8 | 36.1 | 36.6 |
| CPSL | S | ✗ | 90.7 | 55.9 | 89.8 | 43.2 | 86.2 | 34.5 | 31.0 | 39.2 | 45.2 | 43.9 | 86.9 | 46.2 | 85.8 | 64.3 | 32.3 | 89.8 | 50.1 | 65.1 | 40.8 | 38.3 | 50.2 |
| VPT+ELR | S | ✗ | 93.8 | 56.0 | 89.9 | 45.4 | 87.1 | 45.4 | 35.0 | 39.9 | 47.4 | 35.1 | 88.4 | 50.3 | 86.9 | 65.9 | 34.5 | 89.4 | 53.4 | 63.5 | 22.5 | 41.5 | 42.6 |
| GtA | S | ✗ | 130.1 | 57.9 | 89.9 | 46.4 | 88.1 | 48.2 | 40.2 | 42.7 | 47.5 | 35.9 | 88.9 | 51.6 | 88.4 | 66.9 | 36.5 | 88.6 | 53.8 | 63.3 | 36.8 | 37.9 | 48.0 |
| Ours | S | ✗ | 115.4 | 57.5 | 93.0 | 55.7 | 87.1 | 47.9 | 39.5 | 38.0 | 43.2 | 31.5 | 87.1 | 51.2 | 83.3 | 65.4 | 36.0 | 88.7 | 55.5 | 64.7 | 35.3 | 40.5 | 49.4 |
| Source-GtA | M | ✗ | 103.7 | 53.0 | 84.4 | 32.1 | 86.7 | 48.0 | 33.6 | 39.5 | 46.5 | 31.6 | 87.7 | 48.3 | 86.5 | 64.7 | 29.8 | 89.2 | 51.7 | 55.2 | 35.8 | 32.7 | 23.8 |
| CPSL | M | ✗ | 85.2 | 53.1 | 85.5 | 31.2 | 86.1 | 41.8 | 34.2 | 34.8 | 45.2 | 30.0 | 87.2 | 50.0 | 86.7 | 62.9 | 27.1 | 89.1 | 59.5 | 61.7 | 34.5 | 35.1 | 27.1 |
| VPT+ELR | M | ✗ | 89.1 | 55.2 | 86.4 | 34.3 | 87.0 | 46.7 | 34.3 | 41.1 | 49.9 | 33.1 | 88.2 | 50.0 | 88.7 | 66.0 | 28.3 | 90.6 | 63.4 | 62.8 | 34.6 | 35.6 | 27.8 |
| GtA | M | ✗ | 122.3 | 55.4 | 86.4 | 31.1 | 87.6 | 44.6 | 35.2 | 41.7 | 47.8 | 34.1 | 88.8 | 47.7 | 89.0 | 65.9 | 30.9 | 89.5 | 58.0 | 59.5 | 42.9 | 36.5 | 35.9 |
| Ours | M | ✗ | 93.7 | **57.3** | 87.3 | 32.3 | 88.0 | 48.5 | 38.3 | 41.8 | 48.0 | 35.8 | 88.8 | 51.5 | 89.0 | 66.1 | 31.8 | 90.0 | 57.8 | 67.8 | 43.4 | 38.7 | 43.4 |

Table 9: Quantitative evaluations without freezing backbone on SYNTHIA → Cityscapes. Different segmentation architectures: S (Swin-B), M (MiT-B5). FB: whether the backbone is frozen. Params (M): number of trainable parameters. **Bold**: the best results based on different source pre-trained models. Underline: the state-of-the-art results. SSS: Standard Single Source.

| Method | Arch | FB | Params (M) | mIOU$_{16}$(%) | mIoU$_{13}$(%) | road | sidewalk | building | wall* | fence* | pole* | light | sign | veg. | sky | person | rider | car | bus | motor | bike |
|---|---|---|---|---|---|---|---|---|---|---|---|---|---|---|---|---|---|---|---|---|---|
| SSS | S | ✗ | 90.7 | 44.6 | 49.8 | 71.8 | 31.7 | 77.5 | 26.5 | 0.8 | 37.8 | 37.0 | 29.1 | 79.9 | 82.7 | 59.7 | 23.6 | 66.9 | 47.0 | 22.1 | 19.1 |
| CPSL | S | ✗ | 90.7 | 46.0 | 52.0 | 77.5 | 35.8 | 79.2 | 34.7 | 1.4 | 24.4 | 41.6 | 32.5 | 80.3 | 79.0 | 55.3 | 25.7 | 64.3 | 45.5 | 31.3 | 27.4 |
| VPT+ELR | S | ✗ | 93.8 | 47.9 | 53.5 | 74.8 | 35.6 | 79.5 | 30.7 | 1.1 | 40.2 | 47.4 | 35.1 | 88.4 | 84.3 | 63.4 | 27.7 | 66.7 | 51.8 | 28.5 | 26.2 |
| Ours | S | ✗ | 115.4 | **52.7** | 59.2 | 88.0 | 47.0 | 83.9 | 31.5 | 1.2 | 40.8 | 43.2 | 34.2 | 84.0 | 86.0 | 64.7 | 27.1 | 85.6 | 56.1 | 31.7 | 38.4 |
| SSS | M | ✗ | 85.2 | 48.6 | 55.0 | 84.9 | 36.0 | 81.5 | 23.5 | 2.1 | 37.9 | 37.8 | 34.0 | 79.7 | 83.2 | 63.5 | 29.2 | 77.5 | 46.0 | 34.8 | 26.8 |
| CPSL | M | ✗ | 85.2 | 50.8 | 57.9 | 86.1 | 36.3 | 81.9 | 24.2 | 0.7 | 35.2 | 38.7 | 35.5 | 81.7 | 84.6 | 64.6 | 31.9 | 82.6 | 49.2 | 45.3 | 34.2 |
| VPT+ELR | M | ✗ | 89.1 | 51.8 | 58.8 | 86.2 | 37.0 | 83.2 | 21.6 | 3.0 | 39.6 | 40.3 | 38.3 | 82.9 | 84.4 | 65.9 | 34.5 | 81.4 | 50.4 | 45.6 | 35.2 |
| Ours | M | ✗ | 93.7 | **52.7** | 59.4 | 87.9 | 39.9 | 84.1 | 28.0 | 3.4 | 40.1 | 39.7 | 38.5 | 82.9 | 86.5 | 66.6 | 31.7 | 85.8 | 46.8 | 42.4 | 39.5 |
| Source-GtA | S | ✗ | 110.4 | 48.8 | 55.0 | 85.3 | 37.0 | 81.0 | 25.4 | 1.8 | 39.1 | 38.5 | 30.3 | 78.5 | 79.6 | 61.1 | 30.5 | 82.7 | 42.4 | 30.9 | 37.3 |
| CPSL | S | ✗ | 90.7 | 49.0 | 55.9 | 83.4 | 38.1 | 81.3 | 28.5 | 2.8 | 27.2 | 33.6 | 39.4 | 81.1 | 72.8 | 57.6 | 33.9 | 83.0 | 37.7 | 34.3 | 50.0 |
| VPT+ELR | S | ✗ | 93.8 | 52.4 | 59.1 | 87.8 | 41.5 | 83.0 | 27.8 | 2.0 | 40.1 | 43.4 | 33.7 | 80.9 | 82.3 | 64.1 | 33.6 | 85.6 | 48.8 | 39.6 | 44.7 |
| GtA | S | ✗ | 130.1 | 53.1 | 59.7 | 87.1 | 39.8 | 82.9 | 30.6 | 2.7 | 40.4 | 44.0 | 31.9 | 80.9 | 84.1 | 67.9 | 33.5 | 85.1 | 50.2 | 37.3 | 51.8 |
| Ours | S | ✗ | 115.4 | **54.4** | 61.1 | 87.7 | 41.6 | 82.9 | 35.7 | 3.2 | 38.2 | 45.6 | 35.7 | 81.4 | 84.0 | 68.2 | 36.2 | 86.0 | 50.5 | 37.8 | 56.3 |
| Source-GtA | M | ✗ | 103.7 | 50.0 | 56.2 | 84.9 | 42.3 | 82.0 | 29.3 | 2.1 | 38.1 | 35.5 | 34.4 | 82.0 | 86.5 | 61.7 | 30.2 | 81.6 | 44.4 | 32.8 | 32.3 |
| CPSL | M | ✗ | 85.2 | 52.1 | 58.8 | 88.4 | 46.6 | 82.7 | 32.3 | 1.6 | 34.7 | 38.8 | 35.5 | 83.7 | 87.5 | 62.5 | 30.8 | 85.1 | 48.6 | 37.6 | 37.2 |
| VPT+ELR | M | ✗ | 89.1 | 52.7 | 58.9 | 83.6 | 39.3 | 83.0 | 33.8 | 3.1 | 39.9 | 38.1 | 37.8 | 82.4 | 88.9 | 65.0 | 33.1 | 85.4 | 49.2 | 38.6 | 41.6 |
| GtA | M | ✗ | 122.3 | 53.3 | 59.8 | 86.5 | 42.7 | 83.9 | 31.5 | 2.7 | 41.8 | 39.6 | 36.0 | 83.6 | 88.7 | 68.1 | 34.1 | 84.4 | 45.2 | 41.4 | 43.2 |
| Ours | M | ✗ | 93.7 | **54.6** | 60.8 | 87.9 | 46.1 | 84.9 | 39.3 | 3.3 | 39.9 | 37.7 | 39.9 | 84.9 | 89.6 | 67.1 | 32.5 | 86.2 | 50.4 | 38.3 | 45.2 |

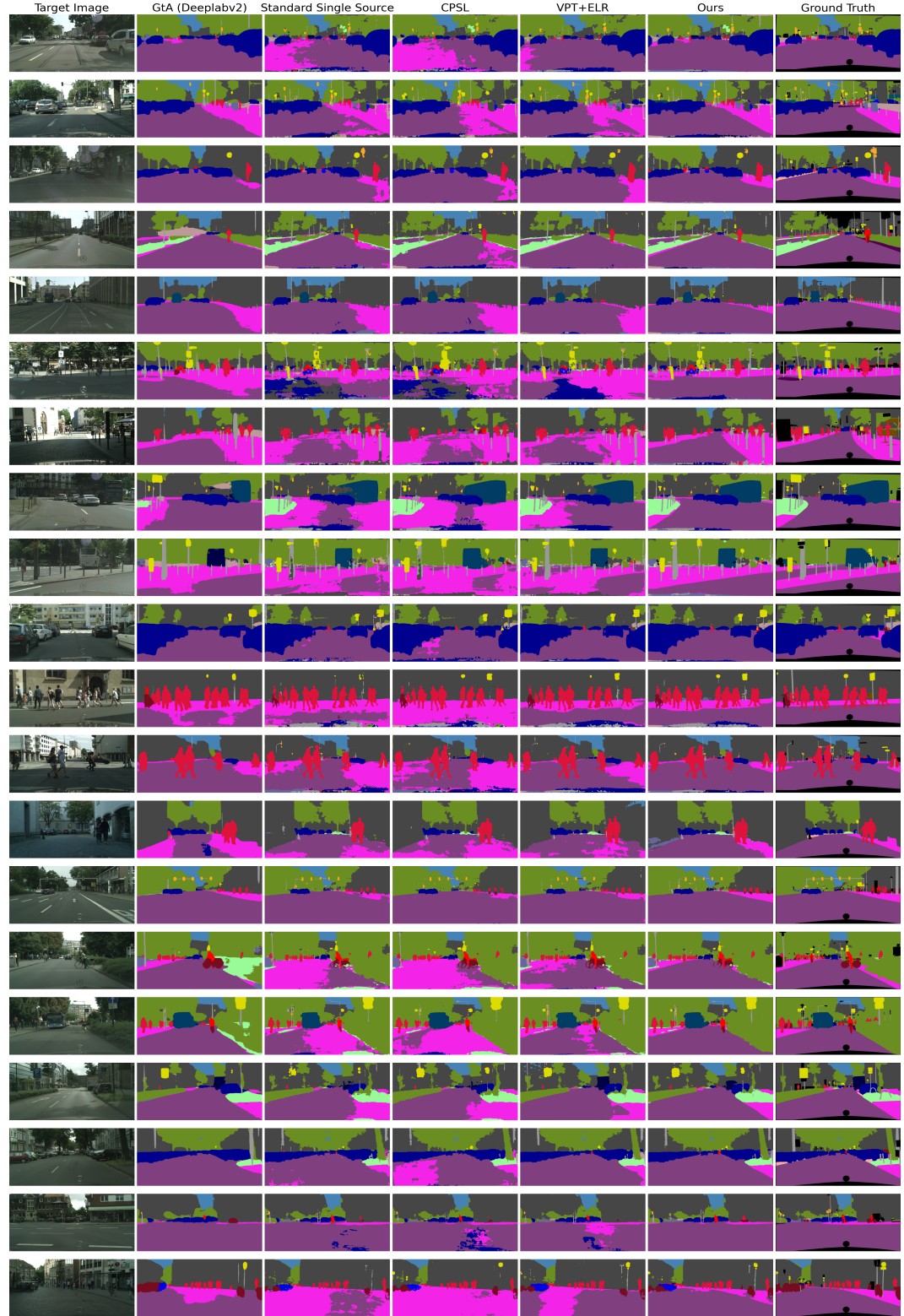

Figure 7: Qualitative results on GTA5 → Cityscapes (Swin, Standard Single Source).

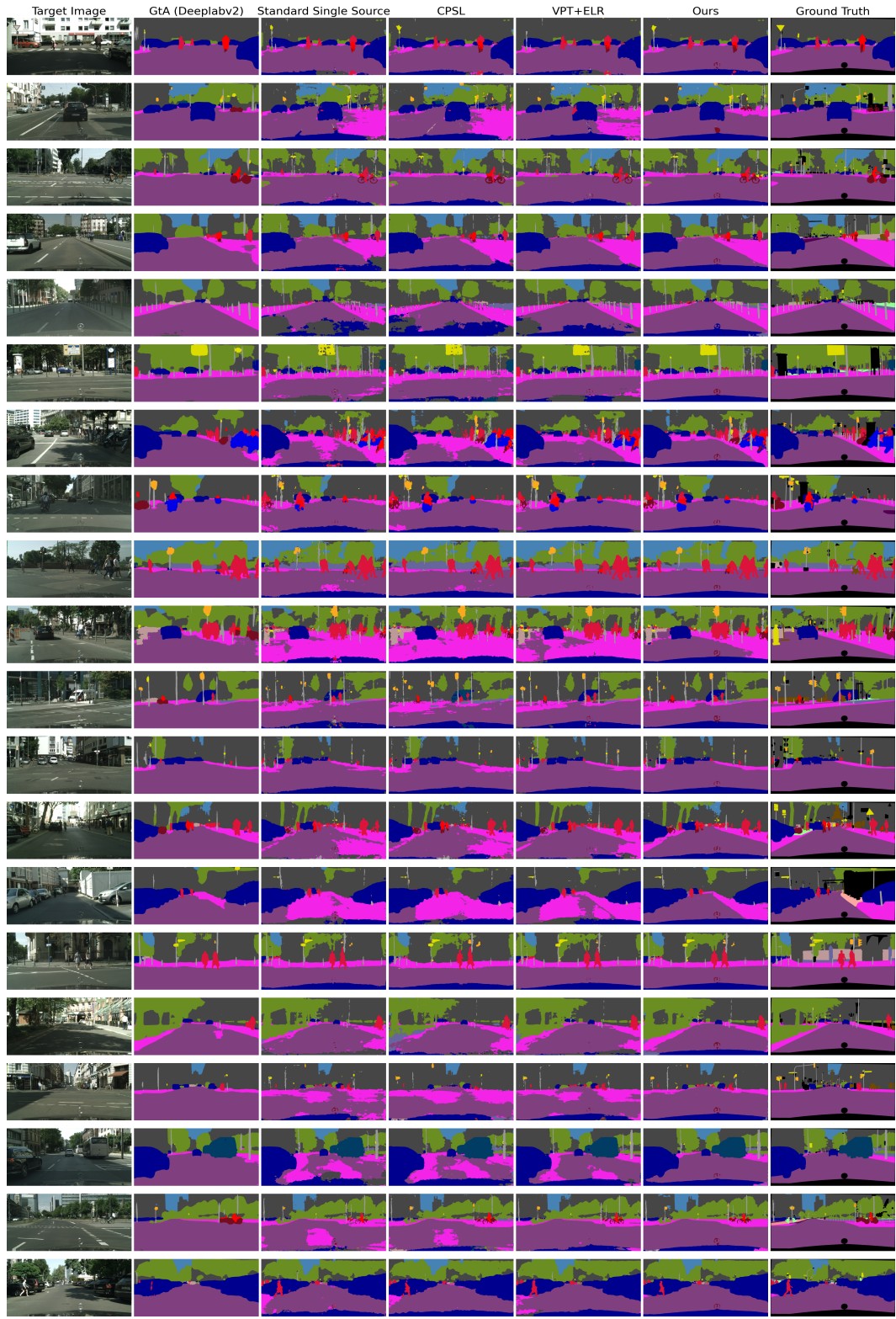

Figure 8: Qualitative results on SYNTHIA → Cityscapes (Swin, Standard Single Source).

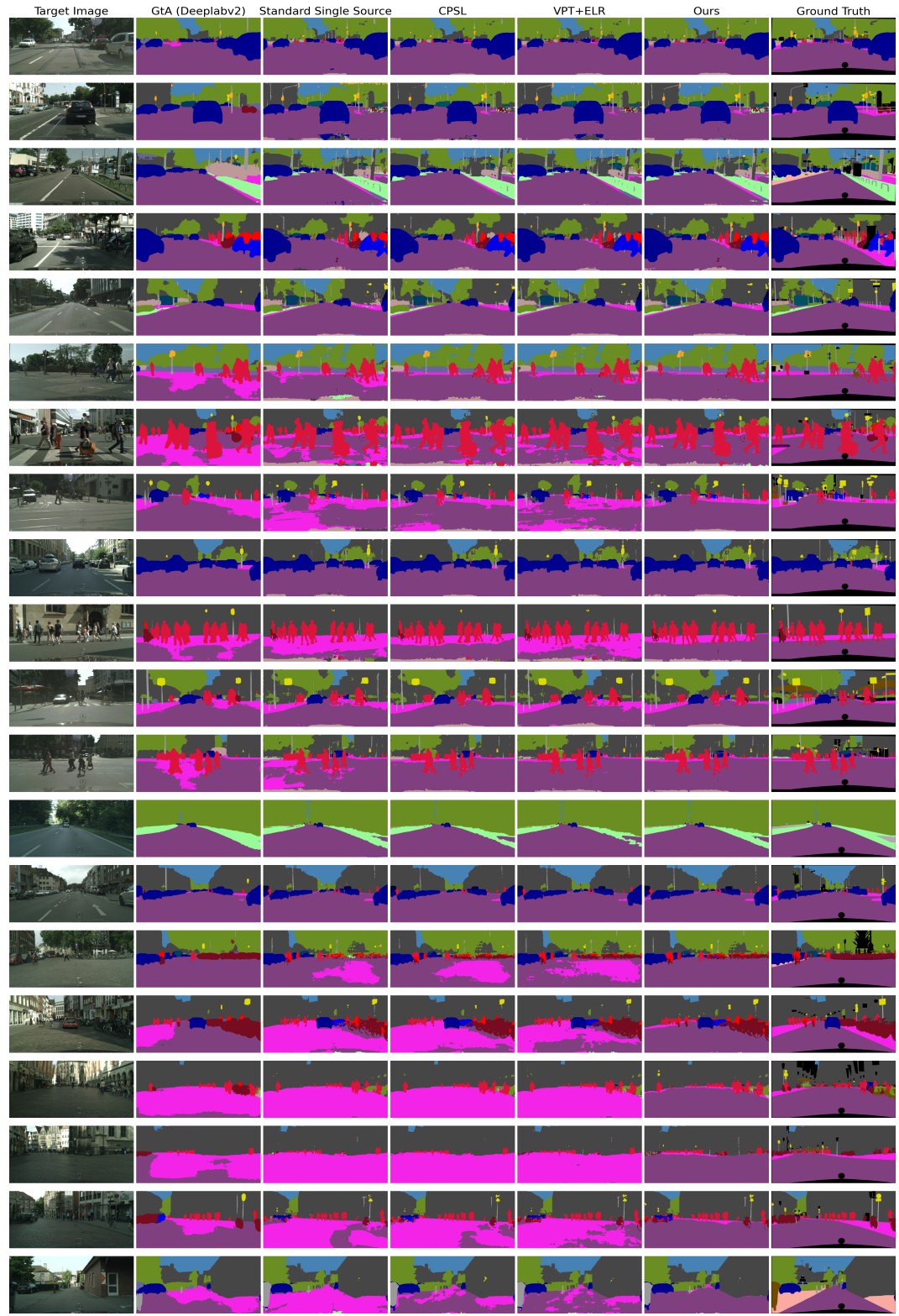

Figure 9: Qualitative results on GTA5 → Cityscapes (MiT-B5, Standard Single Source).

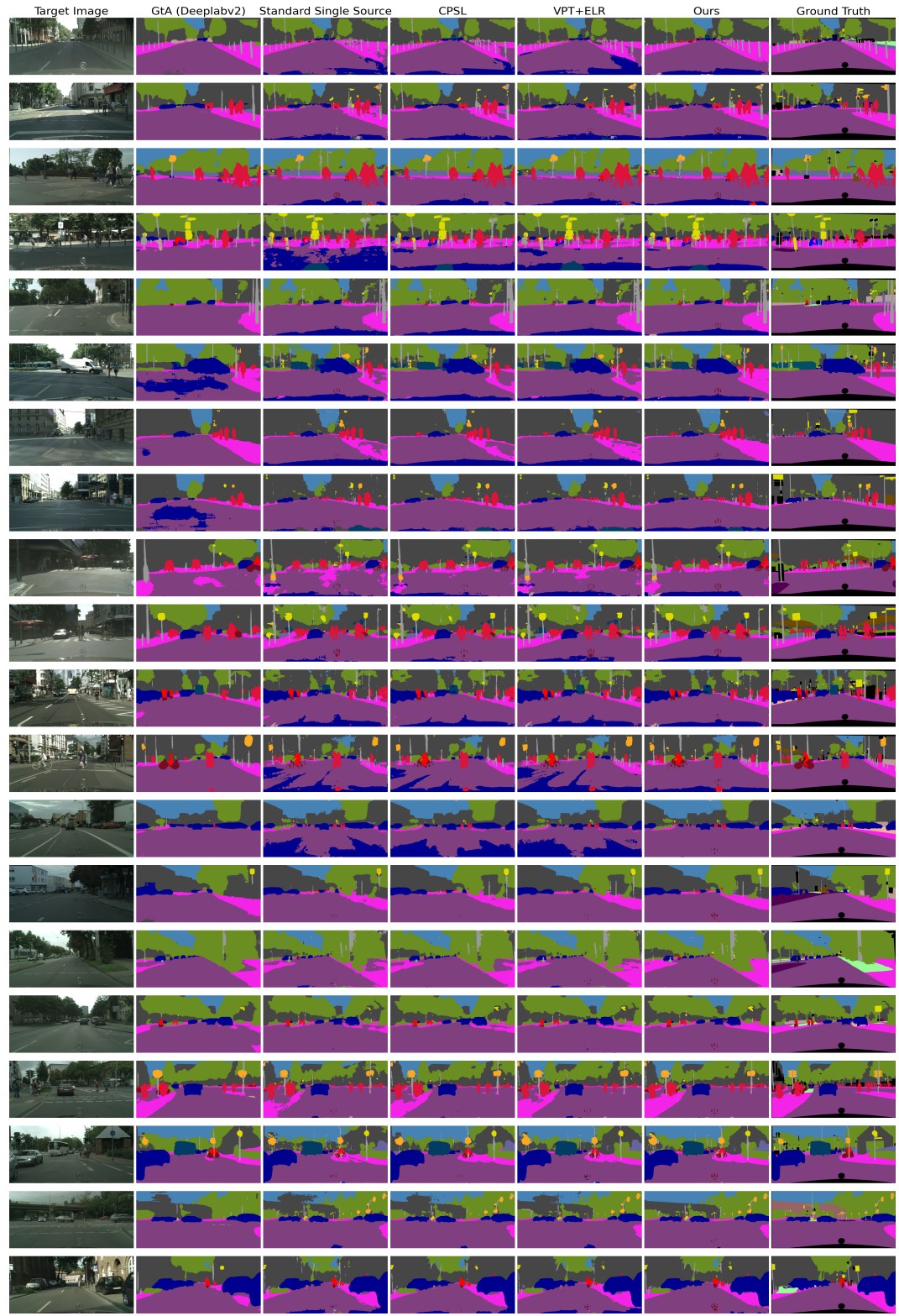

Figure 10: Qualitative results on SYNTHIA → Cityscapes (MiT-B5, Standard Single Source).

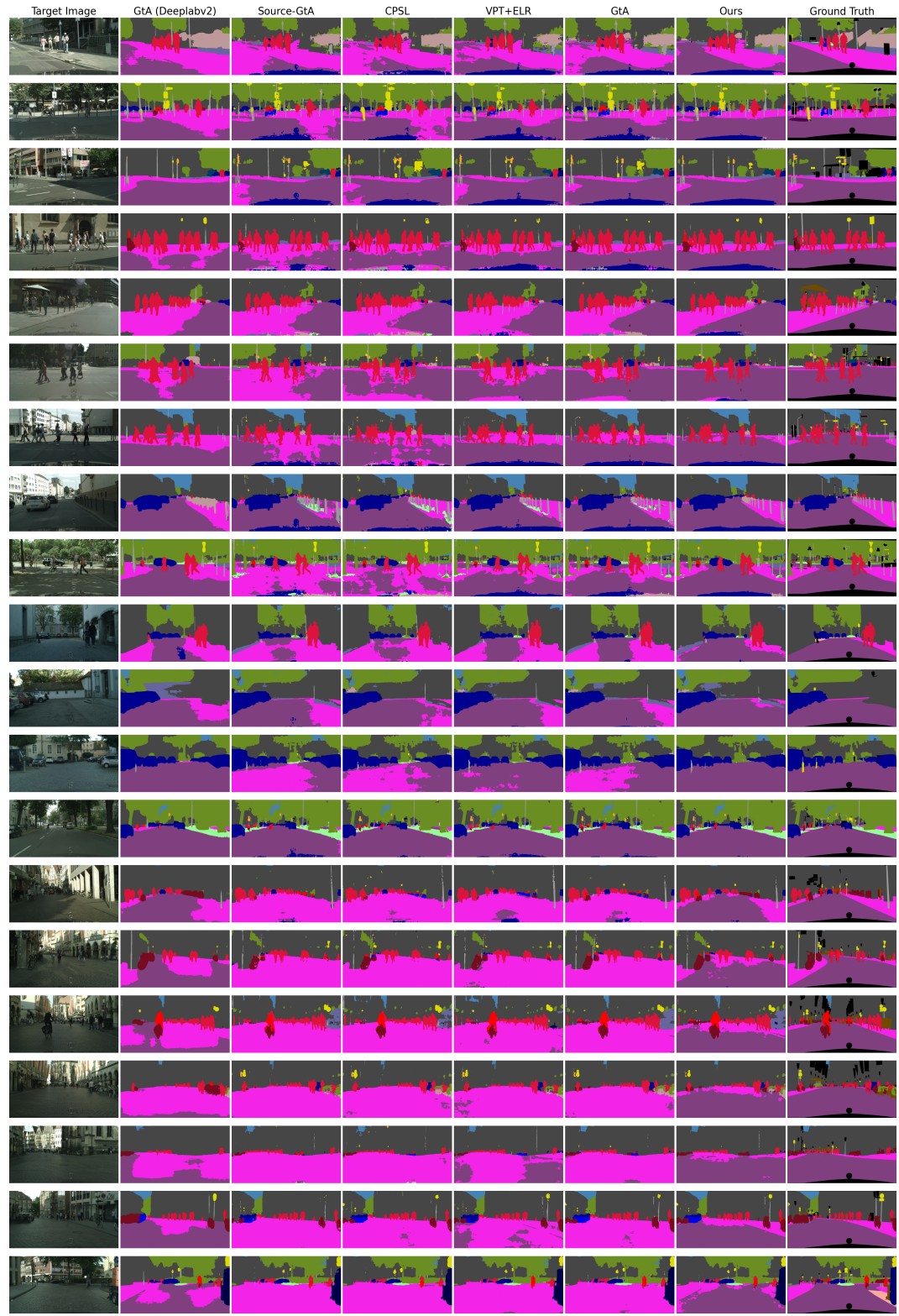

Figure 11: Qualitative results on GTA5 → Cityscapes (Swin, Source-GtA).

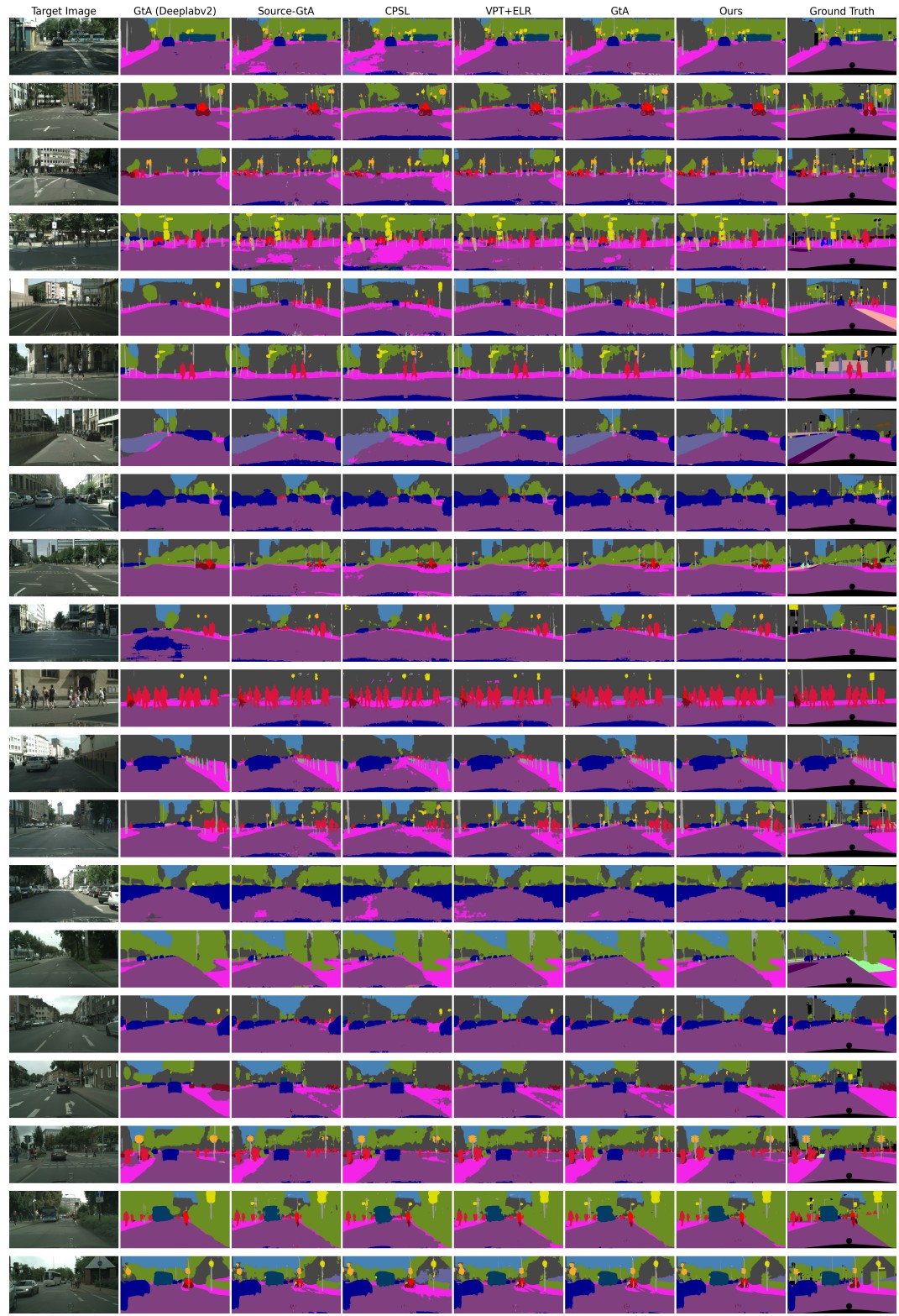

Figure 12: Qualitative results on SYNTHIA → Cityscapes (Swin, Source-GtA).

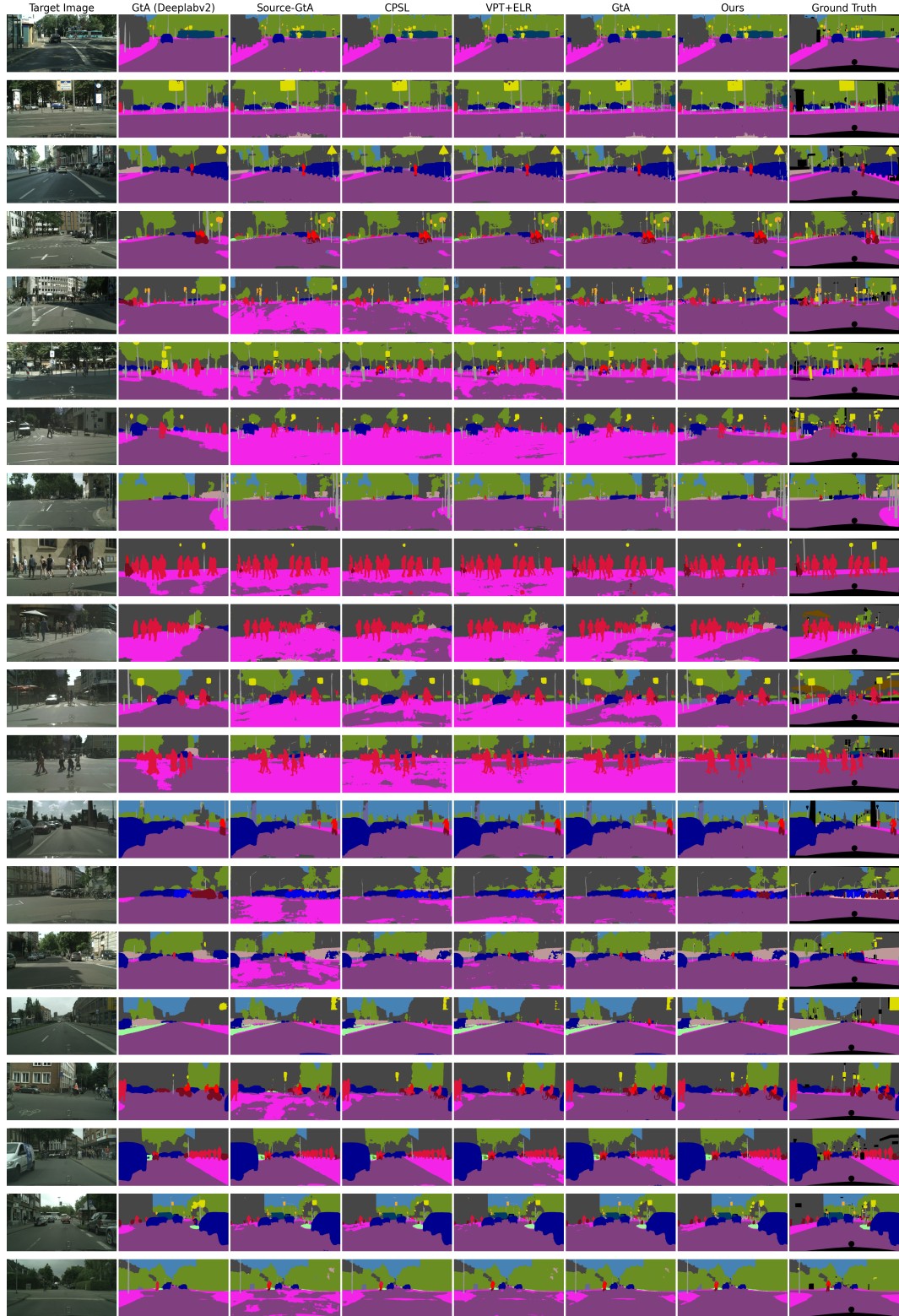

Figure 13: Qualitative results on GTA5 → Cityscapes (MiT-B5, Source-GtA).

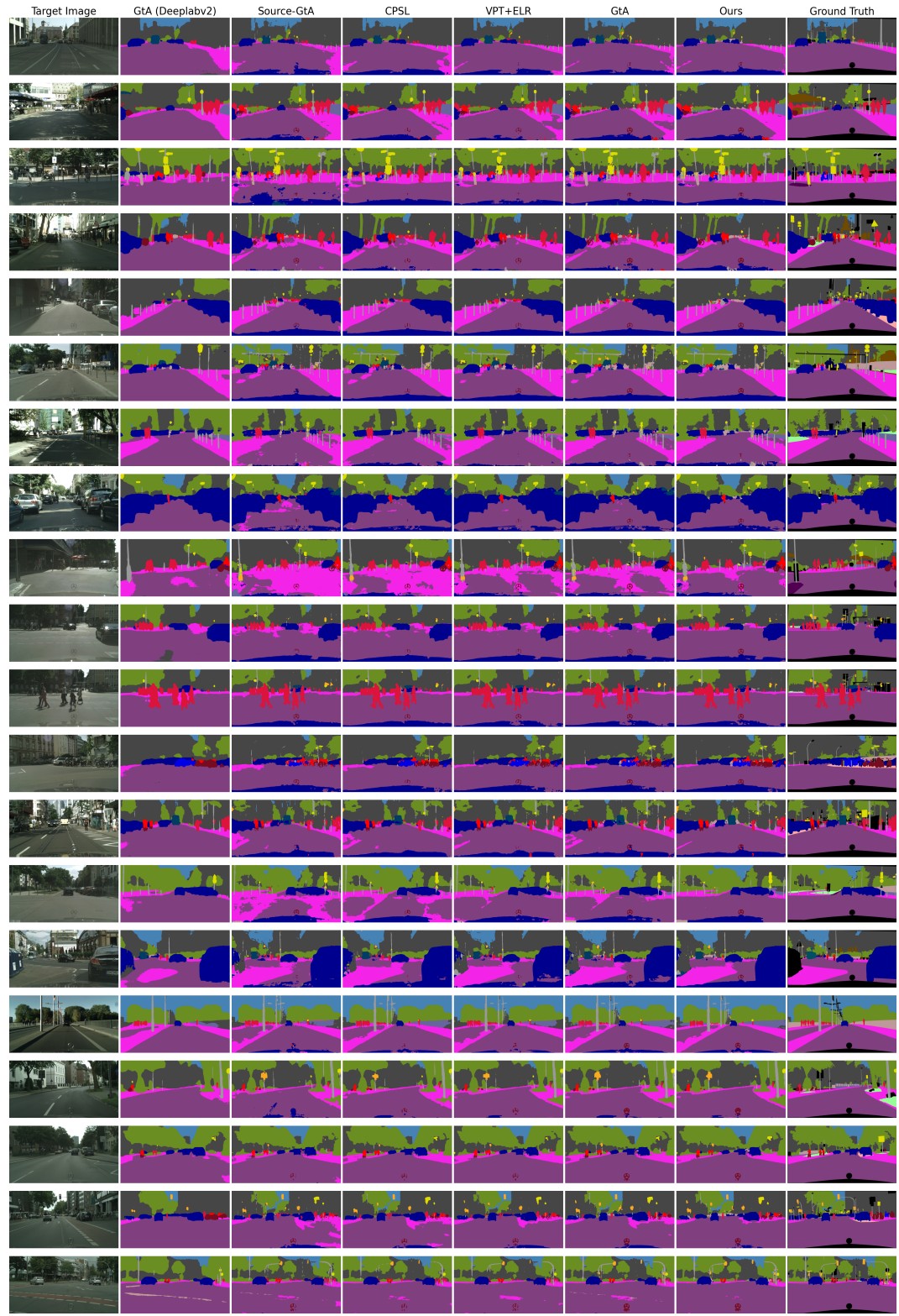

Figure 14: Qualitative results on SYNTHIA → Cityscapes (MiT-B5, Source-GtA).