# OpenReview forum: "When Visual Prompt Tuning Meets Source-Free Domain Adaptive Semantic Segmentation"
_NeurIPS.cc/2023/Conference — NeurIPS 2023 poster_

### Official Review · Reviewer_HKNc · 2023-07-04

**Soundness:** 2 fair
**Presentation:** 3 good
**Contribution:** 2 fair
**Rating:** 4
**Confidence:** 4

**Summary:**

This paper introduces a visual prompt tuning method to tackle source-free domain adaptation in semantic segmentation. The approach involves partitioning the frozen pre-trained model into multiple stages and proposing a prompt adapter for stage-wise prompt tuning.


**Strengths:**

The motivation is clear to apply a prompt tuning method to address the source-free domain adaptation.

The adaptive pixel pruning method is proposed to compute the KL divergence of each pixel and incorporate it into the modified KD loss to get an adaptive threshold which is used to select valuable pixels in each image during training.


**Weaknesses:**

The proposed prompt tuning method is supposed to be parameter-efficient, however, as compared to previous methods, such as CPSL and VPT, the proposed method requires much more leanable parameters. When the backbone is frozen, the prompt adapter requires an additional 2 to 4 times the number of parameters compared to previous methods.

The performance gains in this method can be attributed to the utilization of a stronger vision transformer model. However, a comparison with commonly used convnets is missing.

In the standard single source setting, the enhancement achieved over VPT is minor, ranging approximately from 0.1% to 1% with the MiT backbone, while also entailing nearly twice the number of learnable parameters.


**Questions:**

How and what is the optimal approach for selecting 'r' samples in the memory queue to facilitate adaptive pseudo-label correction?

The learnable vector γ in Eq. (3) is initialized with 0, how does it change during training? How does it affect the training of the prompt interactor?

What is the performance of the proposed prompt tuning method when applied to convolutional neural networks, such as deeplabv2 and fcn, in order to provide a fair comparison with other state-of-the-art methods?


**Limitations:**

How do the proposed prompt tuning methods perform on other SFDA tasks, such as classification?

The comparison experiments are not sufficient, for example the experiment using the same convnets backbone is missing. When using the same vision transformer backbone, the improvement over previous prompt tuning methods is limited.

---

> ### Author Rebuttal · Authors · 2023-08-08
>
> **Q1: The proposed prompt tuning method is supposed to be parameter-efficient, however, as compared to previous methods, such as CPSL and VPT, the proposed method requires much more learnable parameters. When the backbone is frozen, the prompt adapter requires an additional 2 to 4 times the number of parameters compared to previous methods.**
>
> **A1**:
> The reviewer misunderstood the concept of parameter-efficient in our paper. We don't claim our approach is more parameter-efficient than previous visual prompt tuning methods. Instead, we propose that prompt tuning is more parameter-efficient for model adaptation in comparison to previous SFDA methods that fine-tune the entire network.
>
> All three methods (CPSL, VPT, and our approach) freeze the backbone and learn a few parameters. Our approach learns more parameters than the others, but the trainable parameters are less than 1/4 of the backbone, demonstrating that our model is more parameter-efficient than backbone fine-tuning.
>
> **Q2: The performance gains in this method can be attributed to the utilization of a stronger vision transformer model. When using the same vision transformer backbone, the improvement over previous prompt tuning methods is limited. In the standard single source setting, the enhancement achieved over VPT is minor, ranging approximately from 0.1% to 1% with the MiT backbone, while also entailing nearly twice the number of learnable parameters.**
>
> **A2**:
> Vision transformer is not the only reason for performance improvements over CNN-based methods.
> As shown in the following table,
> when the performances of pre-trained CNN and transformer models are similar,
> our performance gains are greater than GtA, illustrating that our approach is important.
>
> |Methods| Arch | GTA5 $\to$ Cityscapes (mIoU19)|
> |:------:|:----:|:----------:|
> |Source-GtA (cited from GtA paper)|D|51.6|
> |GtA |   D  |  53.4(+1.8) |
> |Source-GtA |   S  |  52.8  |
> |Ours|   S  |  56.9 (+4.1) |
> |Source-GtA |   M  |   53.0|
> |Ours|   M  |56.1 (+3.1) |
>
> We respectfully disagree that our improvements are limited/minor.
> As shown in the following table, we calculate our performance improvements over VPT according to Table 1.
> Our approach significantly outperforms VPT in most settings. In the single-source standard setting with Mit backbone, our approach performs comparably.
>
> Setting| Arch |  GTA5 $\to$ Cityscapes (+mIoU19) | SYNTHIA $\to$ Cityscapes (+mIoU16) | SYNTHIA $\to$ Cityscapes (+mIoU13)     |
> |:----:|:----:|:----:|:-------:|------|
> |Standard Single Source|S| +2.7 |   +4.9  | +6.2 |
> |Standard Single Source| M  | +0.1 |   +1.0  | +1.3 |
> |GtA-Source| S | +1.8 |   +2.2  | +2.2 |
> |GtA-Source| M | +1.7 |   +0.8  | +0.6 |
>
> Compared with the source pre-trained model (MiT-B5) containing 85.2M trainable parameters,
> VPT requires 7.6M and ours requires 12.3M, demonstrating that prompt tuning methods significantly reduce learnable parameters.
> Therefore, a few more parameters (less than 5M) in our approach are acceptable for superior performance.
>
> **Q3: A comparison with commonly used convnets is missing. What is the performance of the proposed prompt tuning method when applied to convolutional neural networks, such as deeplabv2 and fcn, in order to provide a fair comparison with other state-of-the-art methods?**
>
> **A3**:
> We have claimed that transferring large-scale source pre-trained models is one of our motivations
> so transformer-based backbones are naturally chosen for our experiments.
> Besides, previous methods do not release the source pre-trained CNN models, making it difficult for fair comparisons.
>
> For a fair comparison with previous CNN-based SFDA methods, we have extended the GtA method with transformer-based backbones.
> Our approach outperforms GtA in both GTA5 $\to$ Cityscape and SYNTHIA $\to$ Cityscape tasks, with a performance improvement ranging from 0.8% to 0.9% and from 0.2% to 1.3% (0.4% to 1.7%), respectively. This shows that our approach is superior to CNN-based SFDA methods.
>
> Due to the limited time, the experiments on CNN models are not finished. The full CNN experiments will be added in the revised paper.
>
>
> **Q4: How and what is the optimal approach for selecting 'r' samples in the memory queue to facilitate adaptive pseudo-label correction?**
>
> **A4**:
> In lines 80-83 of the supplemental material, we have discussed how to choose the suitable memory queue length $r$.
> Specifically,
> we set $r$ as different values and perform complete training.
> The results are shown in Figure 2(c) of the supplemental material.
> We find that higher $r$ leads to better performance but brings more memory cost.
> To balance the memory efficiency and performance, we set r = 1000 in our experiments.
>
> **Q5: The learnable vector γ in Eq. (3) is initialized with 0, how does it change during training? How does it affect the training of the prompt interactor?**
>
> **A5**:
> $\gamma$ is optimized by Adam optimizer with back-propagation gradients, the same as other model parameters.
> Its effects on the prompt interactor are mainly to balance the backbone features $F_{i-1}^{out}$ and the attention layer's output (prompt knowledge), which has been discussed in lines 191-194 of the main manuscript.
> Specifically,
> at the early stage of training, the prompt knowledge is not learned well, and directly injecting prompt knowledge with backbone features may be harmful.
> Therefore, $\gamma$ is initialized with 0 and changes slightly to make $F_i^{in}$ similar to $F_{i-1}^{out}$.
> When the prompt knowledge is well learned, $\gamma$ is learned to adaptively inject prompt knowledge into backbone features.
>
> **Q6: How do the proposed prompt tuning methods perform on other SFDA tasks, such as classification?**
>
> **A6**:
> Theoretically, the proposed prompt tuning methods can be applied to other SFDA tasks.
> Due to the limited time, we cannot report results on the source-free cross-domain classification task. We will add the results in the revised manuscript.

---

> > ### Comment · Reviewer_HKNc · 2023-08-21
> > **Post-rebuttal**
> >
> > Thank you for the detailed clarification provided by the authors.
> >
> > The authors point out that the proposed method is more parameter-efficient than fine-tuning. The novel part is the application of prompt tuning to the domain adaptation task. The authors do this by adopting and verifying prompt tuning methods, such as VPT, and then proposing an incremental method that uses a bit more complex adapter to achieve higher performance. However, it is still unclear whether increasing the number of learnable parameters will always lead to improved performance.
> >
> > In the standard single-source setting, the improvements over VPT with the Swin backbone (first row) are significant, but this comes at a larger number of parameters (**28.6M vs. 7.0M**). When using the MiT backbone and with comparable parameters (**12.3M vs. 7.6M**), the gains are more modest.
> >
> > The authors have extended the GtA method with transformer-based backbones. The improvements over GtA are limited, ranging from **0.2% to 1.7%**. I cannot understand how this can be used to compare the proposed approach to CNN-based SFDA methods, if both are transformer-based? The CNN-based results are missing so far.
> >
> > Therefore, I would keep the score.

---

> > > ### Author Response · Authors · 2023-08-21
> > > **Response to the post-rebuttal**
> > >
> > > We thank the reviewer's comments. We clarify the reviewer's concerns as follows.
> > >
> > > 1. **Differences between VPT and our approach.** Our approach is not an incremental method based on VPT. VPT directly adds learnable tokens (prompts) to the input sequences, while our approach encodes multiscale and task-shared knowledge into prompts and conducts knowledge exchange in the backbone stages.
> > >
> > > 2. **Increasing the number of learnable parameters cannot lead to performance improvements.** For VPT,  we have enlarged its parameters but the performance decreases dramatically due to the ovfitting on pseudo-label noises in the target domain. We have implemented VPT with various parameters and reported the best results in Table 1.  A similar experimental phenomenon in supervised learning tasks can be observed in the original VPT paper (ablation study in prompt length). The performance gains of our approach mainly come from multiscale spatial and task-shared knowledge that helps to align source and target domains.
> > >
> > > 3. **Most trainable parameters in our approach are used for changing feature size, which is not correlated with performance.**  The performance improvements and trainable parameters of our approach are both determined by the feature sizes of the backbones. As the backbone feature sizes of swin-b and mit-b5 are different, the trainable parameters are also different, i.e., 28.6M for swin-b and 12.3M for mit-b5. We cannot arbitrarily change trainable parameters for controlling performance because the feature sizes of backbones are fixed. The performance on mit-b5 is less significant than swin-b due to the smaller feature sizes of mit-b5 than those of swin-b.
> > >
> > > 4. **CNN experiments.** In the original GtA paper (or in Table 1 of our paper), **GtA with CNN backbones** achieves state-of-the-art performances compared with previous CNN-based SFDA methods. It is noticed that one of our motivations is to transfer large-scale models, so our approach is designed for transformer-based backbones. For fair comparison, all previous CNN-based SFDA methods should be extended with transformer-based backbones but most of them do not release their codes. Compromisely,  we extended the GtA with transformer-based backbones for comparison.

---

### Official Review · Reviewer_Ytkx · 2023-07-05

**Soundness:** 3 good
**Presentation:** 3 good
**Contribution:** 3 good
**Rating:** 4
**Confidence:** 5

**Summary:**

This paper considers the problem of source-free domain adaptive semantic segmentation. To solve this problem, the authors propose a visual prompt tuning approach, which keeps the parameters of the networks while adapts the model to target domain only by learning visual prompts. Specifically, a lightweight prompt adapter proposed to learn informative knowledge for segmentation and a label-correction approach is proposed to generate more accurate pseudo-labels. Experiments on two settings show the benefit of the proposed method.

**Strengths:**

+ This paper is well-written and easy to follow. The reviewer can easily understand the motivation of this paper and the method. Experiments are easy to check the benefit of the proposed method.

+ A novel visual prompt tuning approach is proposed. To my knowledge, this is the first prompt tuning work for source-free domain adaptation.

+ A simple label correction is proposed, which is inspired by the phenomenon.

+ Experiments show that the proposed method achieves the best results when fixing the backbone.

**Weaknesses:**

- The benefit of fixing backbone during adaptation is not well verified. 1. The authors state that fine tuning the backbone will cause knowledge forgetting. However, the authors do not conduct experiments that the proposed method could avoid this problem. 2. The computation cost is not discussed. As my understanding, although the proposed method does not need to fune-tune the parameters of the backbone, the computation cost is mostly the same as the backbone fune-tuning methods. This is because the network needs to propagate the gradients to all layers.

- From Table 2, it seems that the proposed PG, PI and LE are not very important. For example, removing all of them only reduces a 0.53 lower mIoU. I thus question that is it required to include them during adaptation which largely increase the complexity.

- The improvement on one setting is very limited, i.e., G-C with M backbone.

- The results on the source domain should be compared as the authors indicate the knowledge forgetting is important.

- As knowledge forgetting is important, it is also required to show the results on continual domain adaptive semantic segmentation task, where there are several target domains sequentially appeared.



**Questions:**

Please see the weaknesses.

**Limitations:**

Yes.

---

> ### Author Rebuttal · Authors · 2023-08-08
>
> **Q1: The computation cost is not discussed. As my understanding, although the proposed method does not need to fune-tune the parameters of the backbone, the computation cost is mostly the same as the backbone fune-tuning methods. This is because the network needs to propagate the gradients to all layers.**
>
> **A1**:
> Our approach is not claimed to be training-efficient but parameter-efficient.
> Therefore, we do not report the computation cost (FLOPS) but the trainable parameters of all compared methods.
>
> We agree that the computation cost of our approach is equal to fine-tuning the backbone,
> but our approach requires less GPU memory and reduces training resources, making it possible for large-scale model adaptation with a single GPU.
>
>
> **Q2: From Table 2, it seems that the proposed PG, PI and LE are not very important. For example, removing all of them only reduces a 0.53 lower mIoU. I thus question that is it required to include them during adaptation which largely increase the complexity.**
>
> **A2**:
> We have revised Table 2 as the table of **@Reviewer zvBk: A1**.
> We are sorry for the performance error of the last line in the table (removing PG, PI, and PE),
> which is revised as 55.07$_{\downarrow 1.17}$.
>
> It is crucial to include PG, PI, and LE modules due to two reasons:\
> (1) As shown in Table 1, the performance gains of previous SOTA methods over source pre-trained models are less than 4%, illustrating these transfer tasks are very challenging.
> Therefore, the performance drops in Table 2 are significant, approximately from 0.66% to 1.17%. \
> (2) The prompt adapter is not complex and is easy to implement, mainly containing convolution layers, trainable vectors, and sparse cross-attention operations.
>
>
> **Q3: The improvement on one setting is very limited, i.e., G-C with M backbone.**
>
> **A3**:
> In the setting of G-C with M backbone,
> our approach achieves comparable performance in the standard single setting
> and outperforms previous SOTA by 0.9% in the Source-GtA setting.
> Please refer to **@Reviewer zvBk: Q2 A2** and **@Reviewer HKNc: Q2 A2** for more discussions.
> It is worth noting that our approach outperforms previous SOTA in most tasks or achieves comparable performance.
>
> **Q4: The authors state that fine-tuning the backbone will cause knowledge forgetting. However, the authors do not conduct experiments that the proposed method could avoid this problem. The results on the source domain should be compared as the authors indicate the knowledge forgetting is important.**
>
> **A4**:
> Dealing with knowledge forgetting is not claimed as our contribution.
> Discussing knowledge forgetting in the paper aims to emphasize that prompt tuning is a better way for model adaptation than fine-tuning the backbone.
> Specifically, prompt tuning methods in NLP/CV usually freeze backbone parameters so that
> the pre-trained knowledge/representations can be utilized for downstream tasks
> while fine-tuning may discard beneficial properties stemming from source pre-training.
> Similar to previous prompt tuning methods, our approach also freezes the backbone to preserve source domain knowledge.
> Finally, we kindly remind the reviewer that our technical contributions are to learn reasonable visual prompts and rectify target pseudo-labels when designing a prompt tuning solution for source-free domain adaptive semantic segmentation.
>
> It is not reasonable to evaluate source domain performance for our approach or other visual prompt tuning methods, due to the following two reasons:\
> (1) Prompts in these methods are designed to reduce the domain gap between pre-trained models and downstream tasks.
> The prompt-related parameters are learned with target data, not source data.
> Therefore, the trained model cannot be used for the source domain.\
> (2) The parameters of the pre-trained backbone are frozen, illustrating source domain knowledge is not forgotten.\
> Therefore, the source domain results of prompt-tuning methods cannot provide valuable insights.
>
>
>
> **Q5: As knowledge forgetting is important, it is also required to show the results on continual domain adaptive semantic segmentation task, where there are several target domains sequentially appeared.**
>
> **A5**:
> Knowledge forgetting is important but not the contribution of this paper.
> In **A4**, we have discussed the reasons in detail.
>
> Continual domain adaptive semantic segmentation is an interesting topic
> but quite different from source-free domain adaptive semantic segmentation.
> The former needs to jointly address knowledge forgetting and domain gap problems while the latter addresses the domain gap.
> Our approach is designed to reduce the domain gap with prompt tuning.
> Therefore, it is unreasonable to apply our approach to the continual domain adaptive semantic segmentation.
>
> Thanks to the reviewer for inspiring us to design prompt tuning approaches for continual domain adaptive semantic segmentation. We will address this interesting topic in our future work.

---

> > ### Comment · Reviewer_Ytkx · 2023-08-20
> > **Thanks for the response**
> >
> > I thank the response from the authors. However, my main concerns were not solved.
> >
> > 1. I understand that the proposed method is parameter-efficient. It is required to show the computational cost compared with other methods to verify the real/actual efficiency. Updating fewer parameters does not equal to efficiency.
> >
> > 2. In the rebuttal, the authors claim that the forgetting is not the contribution of this paper. However, they use the knowledge forgetting as the motivation of prompt tuning in the submission. This will largely mislead the readers (at least me) that this paper is motivated by avoiding knowledge forgetting. I think this is a critical issue, since one main contribution of a paper is guiding the readers to know the motivation of the proposed method and to know which direction is good to solve the problems (e.g., knowledge forgetting) of existing methods. If the authors do not regard knowledge forgetting as a contribution, they should not use knowledge forgetting as a motivation for the proposed method. Specifically, in the abstract, the authors stated **Previous methods usually fine-tune the entire network, which suffers from expensive parameter tuning and unpredictable catastrophic forgetting problems.To avoid these issues, we propose to utilize visual prompt tuning for parameter efficient adaptation.** This statement clearly indicates that the proposed method is introduced to avoid these issues (including the catastrophic forgetting). I thus believe the authors should reconsider the motivation of the proposed method to avoid misleading to the following researchers.
> >
> > Considering the above concerns, I decide to keep my original rating.

---

> > > ### Author Response · Authors · 2023-08-21
> > > **Response to Q1**
> > >
> > > We thank the reviewer's comments. We follow previous visual prompt methods (VPT [1]) and report trainable parameters for "parameter-efficient" comparisons. The computational costs are not compared in previous work. We will add the computational cost in the revised manuscript.
> > >
> > > [1] Jia M, Tang L, Chen B C, et al. Visual prompt tuning, ECCV, 2022: 709-727.

---

> > > ### Author Response · Authors · 2023-08-21
> > > **Response to Q2**
> > >
> > > We thank the reviewer's comments. We must clarify the reviewer's concerns about our motivations for knowledge forgetting.
> > >
> > > Previous fine-tuning-based SFDA methods suffer from expensive parameter tuning and unpredictable catastrophic forgetting problems. Fortunately, prompt-tuning methods in CV/ NLP enable fewer parameters for generalization and freeze pre-trained backbones to avoid knowledge forgetting. **It is our contribution and motivation to point out that prompt-tuning may be better than fine-tuning for source-free domain adaptive semantic segmentation.**
> > >
> > > Similar to previous prompt-tuning methods, our approach inherits the advantages of containing fewer trainable parameters and avoiding knowledge forgetting.
> > >
> > > Thanks for the reviewer's suggestions. We will revise the manuscript for possible misunderstandings.

---

### Official Review · Reviewer_bcKo · 2023-07-05

**Soundness:** 4 excellent
**Presentation:** 3 good
**Contribution:** 3 good
**Rating:** 7
**Confidence:** 5

**Summary:**

The paper presents an unsupervised visual prompting tuning solution for source-free semantic segmentation, where a pretrained model from a source domain is applied to an unlabeled target domain without accessing source data. The authors design a light-weight prompt adapter module to generate knowledgeable visual prompts and adapt target features via cross-attention. Meanwhile, an adaptive pseudo-label correction is proposed to generate reliable pseudo-labels at suitable moments. Experiments on GTA5 $\to$ Cityscapes and SYNTHIA $\to$ Cityscapes tasks validate the efficacy of their method across multiple pretrained models.

**Strengths:**

In general, I like the motivation and technical aspects of this paper. I feel the problem is important and the approach seems to make a step towards the right direction. This paper has a simple and effective idea of the important problem and has great application values. All the technical contributions are well motivated and well ablated. The state-of-the-art comparisons to existing methods are currently a little bit beyond expectation. This paper would be stronger in improving the weakness. Besides, I think the idea of the proposed methods can be extended to CNN backbones and other unlabeled applications except for semantic segmentation. To support my rating, I have listed the strength of this paper,
1. Applying large-scale pre-trained models to unlabeled target domains (with distribution shift) at low cost is an important and meaningful topic. The motivation is well explained and makes sense. The proposed technique introducing visual prompt tuning into source-free DA, is simple yet effective.
2. The overall design of the prompt adapter architecture is quite general and can be used in different applications. Especially, using conv-based prompt generator modules seems to be effective and plays an important role. Unlike the previous visual prompt models, the adapter seems to also make use of the input image which can be effectively leveraged along with intermediate extracted features. In addition, the idea of prompt and features bi-direction refinement between layers in backbones seems to be more optimal than VPT which directly prepends prompts in the input token sequences.
3. The design of the adaptive pseudo-label correction is novel and interesting. Previous methods focus on mining confident pseudo labels via learned feature structures while this paper proposes to correct pseudo labels at several important time moments that are determined by the training IoU curves. Besides, this strategy is simple and universal, which can cooperate with existing pseudo-label strategies and can be applied to other unsupervised and semi-supervised applications.
4. The method has been thoroughly tested on many different source models (different transformer-based backbones pre-trained with different methods) and achieves SOTA performance on GTA5 $\to$ Cityscapes and SYNTHIA $\to$ Cityscapes tasks.
Moreover, the method is conceptually simple and the code has been provided in the supplemental materials for better reproducibility.

**Weaknesses:**

1) Multiscale samples and prompt interactors are important according to Table 2. I wonder whether other augmented samples also work well in the proposed solution.

2) I suggest authors add a figure to show the differences between the proposed method and previous solutions, making the motivations more clear to readers.

**Questions:**

Please refer to the weaknesses.

**Limitations:**

Yes.

---

> ### Author Rebuttal · Authors · 2023-08-08
>
> **Q1: I wonder whether other augmented samples also work well in the proposed solution.**
>
> **A1**:
> To analyze the influences of other augmented samples, we replace scale augmentation with weather augmentation, e.g., snow and frost.
> That is, we replace downscale/original/upscale images with snow/original/frost images for unsupervised visual prompt tuning.
> We summarize the performance of source pre-trained models, Ours and Ours-weather in the following table.
> Ours refers to our approach trained with scale-augmented samples.
> Ours-weather is our approach trained with weather-augmented samples.\
> Compared with Ours,
> Ours-weather improves mIoU approximately -1.5% to -0.1% in GTA $\to$ Cityscapes task
> and -0.1 % to +1.6% in SYNTHIA $\to$ Cityscapes task.
> The experimental results illustrate that
> other augmented samples also work well for unsupervised visual prompt tuning
> and the SOTA results in SYNTHIA $\to$ Cityscapes tasks are updated by weather augmentation.
>
> |         Methods        | Arch | FB | Params (M) | GTA $\to$ Cityscapes (mIoU19) | SYNTHIA $\to$ Cityscapes (mIoU16)|
> |:----------------------:|:---:|:--:|:---------:|:--------:|:---------:|
> | Standard Single Source |   S  |  N |    90.7   | 50.5          | 44.6            |
> |          Ours          |   S  |  Y |    28.6   | 56.2 (+5.7)   | 52.6 (+8.0)     |
> |        Ours-weather    |   S  |  Y |    28.6   | 54.7 (+4.2)   | 52.9 (+8.3)     |
> | Standard Single Source |   M  |  N |    85.2   | 52.5          | 48.6            |
> |          Ours          |   M  |  Y |    12.3   | 54.2 (+1.7)   | 52.6 (+4.0)     |
> |        Ours-weather    |   M  |  Y |    12.3   | 54.1 (+1.6)   | 53.0 (+4.4)     |
> |       Source-GtA       |   S  |  N |   110.4   | 52.8          | 48.8            |
> |          Ours          |   S  |  Y |    28.6   | 56.9 (+4.1)   | 53.8 (+5.0)     |
> |        Ours-weather    |   S  |  Y |    28.6   | 56.3 (+3.5)   | 53.7 (+4.9)     |
> |       Source-GtA       |   M  |  N |   103.7   | 53.0          | 50.0            |
> |          Ours          |   M  |  Y |    12.3   | 56.1 (+3.1)   | 53.8 (+3.8)     |
> |        Ours-weather    |   M  |  Y |    12.3   | 55.2 (+2.2)   | 54.5 (+4.5)     |
>
>
>
> **Q2: I suggest authors add a figure to show the differences between the proposed method and previous solutions, making the motivations more clear to readers.**
>
> **A2**:
> Thanks for the reviewer's suggestion.
> In the revised manuscript, we will add a figure to show the key motivations and differences from previous work.

---

### Official Review · Reviewer_zvBk · 2023-07-07

**Soundness:** 3 good
**Presentation:** 3 good
**Contribution:** 3 good
**Rating:** 6
**Confidence:** 4

**Summary:**

The authors present Uni-UVPT (Universal Unsupervised Visual Prompt Tuning), a framework designed to enable the adaptation of a pre-trained source model, originally trained on a large dataset, for source-free domain adaptive semantic segmentation. Their approach involves introducing a unique prompt adapter that gradually integrates relevant knowledge into prompts, facilitating alignment between the target features and the pre-trained model. They also propose an adaptive strategy for correcting pseudo-labels, along with a multi-scale consistency loss, which enhances the spatial reliability of visual prompts by rectifying pseudo-labels at suitable instances.

**Strengths:**

The paper's structure is well-organized, and it effectively addresses the problem at hand while clearly outlining its contributions.
The inclusion of experiments using models with different backbones is valuable for analyzing diversity and understanding the results obtained through the experiments.

**Weaknesses:**

It would be beneficial to include a parameter comparison not only with the previous method but also among the proposed methods in the ablation study.  This would allow us to determine which specific component methods have the most significant impact on memory efficiency.

**Questions:**

Based on the findings presented in Table 1, the experimental results for the MiT-B5 (M) backbone do not exhibit a noticeable distinction in performance when compared to the results of Ours and GtA.

**Limitations:**

They mentioned limitation of proposed method.

---

> ### Author Rebuttal · Authors · 2023-08-08
>
> **Q1: It would be beneficial to include a parameter comparison not only with the previous method but also among the proposed methods in the ablation study. This would allow us to determine which specific component methods have the most significant impact on memory efficiency.**
>
> **A1**:
> Thanks for the reviewer's suggestions.
> We have provided parameter comparisons of previous methods in Table 1.
> Besides, the parameter comparisons for the ablation study
> are shown in the following table (the revised Table 2).
>
> | Stem        | LE | PI   | Params(M) | mIoU(%)      |
> |:-------------:|:----:|:------:|:-----------:|:--------------:|
> | Multiscale  | &check;  | &check; | 28.6081  | 56.24 (Ours) |
> | Multiscale  | &cross;  | &check; | 28.6077  | 55.58 $_{\downarrow 0.66}$ |
> | Singlescale | &check;  | &check; | 15.4827  | 55.52 $_{\downarrow 0.72}$ |
> | &cross;     | &check;  | &check; | 27.5138  | 55.50 $_{\downarrow 0.74}$ |
> | Multiscale  | &check;  | PI 1    | 6.0624   | 55.34 $_{\downarrow 0.90}$ |
> | &cross;     | &cross;  | &cross; | 3.9417   | 55.07 $_{\downarrow 1.17}$ |
>
> From the results, we can calculate the parameters of all modules.
> Specifically, Level Embedding (LE) and Stem are respectively 4e-4M and 1.09M.
> The head contains less than 4M parameters.
> Most of the trainable parameters in the full model belong to prompt interactors (PG), i.e., 23.57M.
> Furthermore, we calculate the parameters of four PG modules, namely, 1.03M, 3.90M, 15.21M, and 3.43M,
> which are positively correlated with the middle features' dimensions of backbone stages.
>
>
>
>
> **Q2: Based on the findings presented in Table 1, the experimental results for the MiT-B5 (M) backbone do not exhibit a noticeable distinction in performance when compared to the results of Ours and GtA.**
>
> **A2**:
> Comparing the results for the MiT-B5 (M) backbone in Table 1, our approach outperforms GtA by 0.9% in GtA5 $\to$ Cityscapes, and 0.2% and 0.4% in SYNTHIA $\to$ Cityscapes.
> Compared with the noticeable performance of the Swin-B backbone,
> the performance improvements for MiT-B5 are smaller due to the following two reasons:\
> (1) To match the middle features' shape of Mit-B5 backbone stages,
> the prompts' dimensions in Mit-B5 are smaller than those in Swin-B,
> which weakens the visual prompts' learning capacity. \
> (2) Following most previous methods, we apply single one segmentation head for predictions,
> while GtA utilizes five auxiliary heads (with more trainable parameters) for consistent predictions.
> If our approach cooperates with the tricks of GtA, the performance of our approach could be further improved but the trainable parameters increase.\
> In summary, our approach is theoretically applicable for all transformer backbones and obtains different performance improvements.

---

> > ### Comment · Reviewer_zvBk · 2023-08-21
> > **Response to Rebuttal**
> >
> > After going through the feedback and the authors' reply, I still think the paper should be accepted. Some might say it's not very new in its methods, but the improvements shown and the fresh approach to using a prompt adapter make it worth publishing.

---

### Author Rebuttal · Authors · 2023-08-08

We thank all reviewers for their valuable comments.
We are encouraged that they find our idea to be unique / simple and effective / great application values / important and meaningful / novel and interesting / the first prompt tuning work for source-free domain adaptation (**Reviewer zvBk, bcKo, Ytkx**),
and our paper is well-organized / well motivated / well-written and easy to follow / clear (**all reviewers**).
We are glad that they think our experiments are valuable for analyzing diversity and understanding the results / validate the efficacy / are well-ablated / are a little bit beyond expectation / achieve the best results when fixing the backbone (**Reviewer zvBk, bcKo, Ytkx**).

We have provided separate responses to all reviewers’ questions.

---

### Decision · Program_Chairs · 2023-09-21

**Decision:**

Accept (poster)

**Comment:**

Source free domain adaptation is a  challenging problem, and authors are trying to solve it using visual prompt tuning.
Ytkx has raised valid point regarding experiment lacking w.r.t. "catastrophic forgetting problems". Authors stated that they do not claim contribution around "knowledge forgetting". Meta-reviewers believe that authors should update draft to make this clear. Especially update following lines
"Previous methods usually fine-tune the entire network, which suffers from expensive parameter tuning and unpredictable catastrophic forgetting problems.  To avoid these issues, we propose to utilize visual prompt tuning for parameter efficient adaptation. "

Please incorporate other points raised by the reviewers too.